# Balanced Twins: Causal Inference on Time Series with Hidden Confounding

## Abstract

Accurately estimating treatment effects in time series is essential for evaluating interventions in real-world applications, especially when treatment assignment is biased by unobserved factors. In many practical settings, interventions are adopted at different times across individuals, leading to staggered treatment exposure and heterogeneous pre-treatment histories. In such cases, aggregating outcome trajectories across treated units is ill-defined, making individual treatment effect (ITE) estimation a prerequisite for reliable causal inference. We therefore study the problem of estimating the average treatment effect for the treated (ATT) by first recovering individual-level counterfactuals. We introduce a neural framework that learns simultaneously low-dimensional latent representations of individual time series and propensity scores. These estimates are then used to approximate the individual treatment effects through a flexible matching procedure that avoids classical convexity constraints commonly used in synthetic control methods. By operating at the individual level, our approach naturally accommodates staggered interventions and improves counterfactual estimation under latent bias, without relying on explicit temporal modeling assumptions. We illustrate our approach on both real-world energy consumption data and clinical time series, including high-frequency electricity demand-response programs and semi-synthetic data for individuals in intensive care unit (ICU), where hidden confounding, staggered treatment adoption, and non-stationary dynamics are prevalent.

## 1 Introduction

Balancing electricity demand and supply has become an increasing challenge for modern energy grids, especially during peak consumption periods. Demand-Response (DR) programs are designed to reduce or shift household electricity usage to alleviate grid stress. However, evaluating their causal impact is difficult due to the complexity of smart meter data and the influence of hidden confounders, such as lifestyle, ecological awareness, and financial constraints, which affect both participation and energy usage.

A significant source of bias arises from the voluntary nature of DR program enrollment. Participants often differ systematically from non-participants, leading to selection bias. Additionally, participation occurs at staggered times among individuals, complicating the estimation of treatment effects due to timing variations. Standard causal inference methods, including matching, regression adjustment, and time series models, typically rely on strong assumptions, such as the absence of unmeasured confounding or parallel trends. Unfortunately, these assumptions are often violated in real-world scenarios.

A further challenge is that outcome dynamics are often *non-stationary*: the relationship between past and future behavior may change over time. In electricity consumption, such changes may occur *at* treatment time, *after* treatment time, or independently of treatment timing. For example, participation in a demand-response program may coincide with a behavioral adaptation, such as shifting appliance usage or changing heating schedules. More generally, consumption patterns may evolve later due to seasonal transitions, temperature changes, policy signals, or the adoption of new equipment such as electric vehicles, heat pumps, or solar panels. These changes are not necessarily synchronized across individuals and may depend on latent factors such as flexibility, financial capacity, or ecological awareness. As a result, two households with similar pre-treatment consumption can exhibit very different post-treatment counterfactual trajectories. In such settings, methods that rely on extrapolating future outcomes from past observations may produce

biased or unstable counterfactual estimates. Similar issues arise in healthcare, where patient trajectories may change because of disease progression, changes in monitoring intensity, ICU transfer, or treatment-associated care conditions, so that post-treatment outcomes no longer follow the same regime as pre-treatment observations.

To address these challenges, we propose a neural framework, Balanced-Twin (B-Twin), for estimating individual treatment effects (ITE) from time series data while accounting for hidden confounders. Our objective is to aggregate these ITEs to accurately determine the average treatment effect for the treated (ATT). The key idea is to learn low-dimensional latent representations of individuals from pre-treatment outcomes and jointly estimate individual propensity scores. These quantities are then used to construct synthetic counterfactuals through a flexible matching procedure, avoiding direct extrapolation of post-treatment dynamics from pre-treatment data.

Importantly, our approach is explicitly grounded in the *synthetic control framework*. Like classical synthetic control methods, B-Twin produces counterfactual outcomes as weighted combinations of observed units, preserving a clear parametric structure and interpretability. Unlike black-box neural approaches that directly regress outcomes or treatment effects, our method yields explicit balancing weights that quantify each control unit's contribution to the counterfactual, making the estimation process transparent and inspectable. At the same time, by learning the matching structure in a latent space, B-Twin relaxes restrictive convexity constraints and remains effective in high-dimensional and noisy settings.

Beyond interpretability, our method is computationally efficient and scalable. By leveraging batch-based weight estimation, B-Twin can handle large datasets without requiring expensive per-unit optimization, a limitation of many synthetic control variants.

We evaluate our approach through extensive experiments on synthetic, semi-synthetic, and real-world datasets from both electricity consumption and healthcare domains. Our results show that classical methods such as synthetic control (Abadie et al., 2010) and SyncTwin (Qian et al., 2021) struggle in the presence of noise and high-dimensional individual data, while neural black-box approaches such as DragonNet (Shi et al., 2019) are sensitive to temporal shifts in the data-generating process. In contrast, B-Twin provides more stable and accurate counterfactual estimates in settings where hidden confounding and non-stationary dynamics make extrapolation unreliable. In summary, our contributions are threefold: (1) an interpretable synthetic-control-inspired framework that jointly learns latent representations and propensity scores for robust ATT estimation under hidden confounding and non stationary dynamics; (2) extensive empirical validation on challenging and realistic benchmarks; and (3) practical insights into the limitations of existing causal inference methods when treatment assignment is influenced by unobserved factors and outcomes extrapolation may be unreliable.

## 2 Related Work

Estimating treatment effects from observational data is a central problem in causal inference, particularly when outcomes are time-dependent and treatment assignment is influenced by unobserved confounders. Existing approaches can be broadly categorized into traditional econometric methods, learning-based models, and recent temporal or latent-variable frameworks.

### 2.1 Traditional Methods

Classical econometric techniques such as Difference-in-Differences (DiD) (Angrist & Pischke 2009; Ashenfelter & Card 1984; Card 1990; Bertrand et al. 2004; Abadie 2005) are widely used for policy evaluation in panel data settings. However, they rely on the strong parallel trends assumption, which is often violated in heterogeneous and noisy real-world environments (Gibson & Zimmerman, 2021). Ordinary Least Squares (OLS) regression (Rubin, 1974; Angrist & Pischke, 2009) is simple and interpretable but assumes that all confounders are observed, making it vulnerable to hidden bias. Propensity Score Matching (PSM) (Caliendo & Kopeinig, 2008) aims to balance treated and control units but similarly assumes no hidden confounding and sufficient overlap.

Synthetic Control Methods (SCM) (Abadie & Gardeazabal 2003; Abadie et al. 2010; Klößner et al. 2018; Abadie & L'hour 2021) extend matching by constructing weighted combinations of control units to approximate counterfactual outcomes. While SCM can implicitly account for latent confounders under strong factor-model assumptions, it is primarily designed for aggregate units and becomes computationally expensive and unstable at the individual level,

especially with high-frequency time series. Synthetic Difference-in-Differences (SDID) (Arkhangelsky et al., 2021; Clarke et al., 2023) improves balance across both units and time, but remains sensitive to numerical instability and scaling issues when applied to large populations with fine-grained temporal data.

## 2.2 Learning-Based Approaches

To overcome the limitations of classical methods, several neural approaches have been proposed for treatment effect estimation. Treatment-Agnostic Representation Networks (TARNet) and Counterfactual Regression Networks (CFRNet) (Shalit et al., 2017) learn representations to predict potential outcomes, with CFRNet enforcing balance between treated and control groups. Balancing Neural Networks (BNN) (Johansson et al., 2016) extend this idea using discrepancy-based regularization. However, these methods assume that all confounders are observed and are not designed to address hidden confounding.

DragonNet (Shi et al., 2019) and BCAUSS (Tesei et al., 2023) augment TARNet with an explicit propensity score estimation head to better structure the latent space, with BCAUSS further addressing violations of the positivity assumption. Despite their effectiveness in cross-sectional settings with observed covariates, these models remain ill-suited for our problem. When applied to time series data, they typically generate counterfactuals by directly regressing outcomes on treatment indicators and pre-treatment trajectories, resulting in black-box predictors that are difficult to interpret and sensitive to non-stationarity in the data-generating process. In contrast, our goal is to remain within a synthetic control–style framework, where treatment effects are identified through explicit weighting schemes that can be directly inspected and understood by domain experts.

Causal Effect Variational Autoencoder (CEVAE) (Louizos et al., 2017) addresses hidden confounding by leveraging proxy covariates within a latent-variable framework, while GANITE (Yoon et al., 2018) uses adversarial training to estimate individualized treatment effects under multiple treatments. Both approaches, however, rely on the availability of informative covariates or proxies and are not tailored to high-frequency time series settings where confounding information is embedded in outcome dynamics rather than observed features.

## 2.3 Temporal and Latent Models

Recent work has focused on treatment effect estimation under temporal confounding, where treatment assignment evolves over time like in (Bica et al. 2020; Liu et al. 2020; Cao et al. 2023). These methods address dynamic and multiple treatment regimes, which introduce additional complexity. In contrast, our setting considers a point intervention with staggered adoption, avoiding temporal confounding while still posing challenges due to hidden confounders and non-stationary outcomes. As a result, dynamic treatment models may be unnecessarily complex for our problem.

SyncTwin (Qian et al., 2021) is closely related to our work and proposes learning latent temporal embeddings to generate synthetic counterfactuals under staggered interventions. While effective for low-frequency longitudinal data, such as clinical records, its reliance on LSTM-based architectures limits scalability and stability in high-dimensional, noisy time series, such as smart meter data.

## 2.4 Our Contribution

We propose a two-phase framework that bridges classical synthetic control methods and neural representation learning for treatment effect estimation under hidden confounding. First, we learn low-dimensional latent representations from pre-treatment outcome trajectories and treatment assignment, under the assumption that treatment-relevant hidden confounders leave recoverable signatures in pre-treatment dynamics. These representations serve as proxies for unobserved confounders and enable individual-level propensity score estimation. Second, we construct individual synthetic controls using a novel balancing condition that generalizes classical convexity constraints. This design preserves interpretability through explicit weighting while achieving robustness and scalability in high-frequency, non stationary, individual-level time series settings where existing methods may fail.

## 3 Problem Formulation

We aim to estimate the *average treatment effect for the treated (ATT)* in an observational setting where individual treatment effects (ITE) vary over time and are influenced by latent factors. Specifically, we consider the estimation of the effect of a treatment $T$ (e.g., participation in an intervention) on an outcome $Y$ (e.g., electricity consumption), in the presence of hidden confounders $W$ (e.g., ecological awareness or socioeconomic status). Outcome trajectories are high-frequency time series, and treatment assignment depends on latent confounders $W$ and may be staggered across individuals. Additional observable covariates $X$, which may include observed confounders, can also be incorporated but are not required in our framework.

We consider a time horizon of length $L$, i.e. $t \in \{1, \dots, L\}$. Let $\mathcal{T}$ denote the treated population and $\mathcal{C}$ the control population. To each individual $i \in \mathcal{T} \cup \mathcal{C}$, we associate observed covariates $X_i = \{X_{i,t}\}_{t=1}^L$ with $X_{i,t} \in \mathbb{R}^D$, hidden confounders $W_i \in \mathbb{R}^K$, outcomes $Y_i = \{Y_{i,t}\}_{t=1}^L$ with $Y_{i,t} \in \mathbb{R}$, and a binary treatment indicator $T_i \in \{0, 1\}$ indicating whether the individual receives treatment during the observation period, with a single adoption time per unit.

Let $Y_i^1$ and $Y_i^0$ denote the potential outcome trajectories of individual $i$ under treatment and control, respectively. More generally, treatment adoption may occur at different times $t_i$ across individuals, leading to a staggered treatment setting. However, for simplicity and to facilitate comparison with existing baselines, all results in this paper consider a common treatment onset time $t_0$. Our primary objective is to estimate accurate individual-level counterfactuals, which can then be aggregated to estimate the ATT even when treatment adoption times differ across individuals. The extension of the framework to staggered treatment adoption is detailed in Appendix A.

Throughout the paper, $(Y^0, Y^1, Y, W, T, X)$ denote generic random variables, while $(Y_i^0, Y_i^1, Y_i, W_i, T_i, X_i)$ denote unit-specific random variables associated with individual $i$.

The goal is to estimate the average treatment effect for the treated:

$$\text{ATT} = \mathbb{E}\left[ \frac{1}{L - t_0 + 1} \sum_{t=t_0}^L \left(Y_t^1 - Y_t^0\right) \,\middle|\, T = 1 \right]$$

$$= \mathbb{E}\left[ \frac{1}{L - t_0 + 1} \sum_{t=t_0}^L \left(Y_t - Y_t^0\right) \,\middle|\, T = 1 \right].$$

In traditional causal inference, when confounders $X$ are observed, identification typically relies on the Strong Ignorability assumption (Rosenbaum & Rubin, 1983), which requires both Positivity ($0 < P(T = 1|X) < 1$) and Ignorability:

$$(Y^1, Y^0) \perp\!\!\!\perp T \mid X.$$

However, in our setting, the confounders $W$ are unobserved, and therefore the standard ignorability assumption does not hold. These hidden factors (e.g., ecological concern, financial status) affect both the likelihood of treatment and the outcome, violating the standard ignorability assumption and introducing bias in naive estimators (Cai & Kuroki, 2012). As a result, observed covariates $X$ are omitted from the experiments and theoretical development for simplicity, as our framework focuses on settings where relevant confounding information is not explicitly observed and must be approximated from outcome trajectories. When available, $X$ can be incorporated alongside outcomes within our framework to mitigate their induced bias.

Specifically, we suppose that treatment adoption depends on W through:

$$g(W) = \mathbb{P}(T = 1|W),$$

and outcome trajectories $Y$ depend on $W$ as shown in Figure 1. Our work relies on the following assumptions:

**Assumption 1.** *(The Stable Unit Treatment Value Assumption) The potential outcomes for one individual are unaffected by the treatment of others.*

**Assumption 2.** *(Positivity) Every individual has a non-zero probability of receiving treatment and control given the latent factor $W$:*

$$0 < \mathbb{P}(T = 1 \mid W = w) < 1.$$

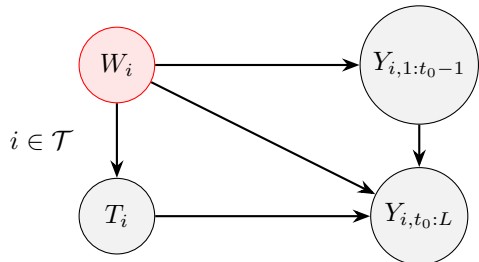

Figure 1: Causal DAG (Joffe et al., 2012) illustrating latent confounders $W_i$, treatment $T_i$, pre-treatment outcome $Y_{i,1:t_0-1}$ and post-treatment outcome $Y_{i,t_0:L}$ in the common treatment-time setting.

**Assumption 3.** *(Latent Ignorability) The potential outcomes are independent of treatment given hidden factors $W$:*

$$(Y^1, Y^0) \perp\!\!\!\perp T \mid W.$$

**Assumption 4.** *(Latent recoverability assumption) Let $Y^{\mathrm{pre}} = \{Y_t\}_{t=1}^{t_0-1}$ denote the pre-treatment outcome trajectory. We assume that there exists a function $h$ and a sufficiently small constant $\varepsilon_W > 0$ such that*

$$\|W - h(Y^{\mathrm{pre}})\| \leq \varepsilon_W.$$

*This means that the pre-treatment trajectory contains sufficient information to approximately recover the treatment-relevant latent factors $W$, up to recovery error $\varepsilon_W$.*

**Assumption 5** (Lipschitz regularity). *The recovery map $h$ and the propensity function $g$ are Lipschitz continuous. That is, for all $y, y'$ and $w, w'$,*

$$\|h(y) - h(y')\| \leq L_h \|y - y'\|,$$

*and*

$$|g(w) - g(w')| \leq L_g \|w - w'\|.$$

The latent recoverability assumption is motivated by the fact that latent factors such as lifestyle, behavioral flexibility, socioeconomic status, or disease severity often induce persistent patterns in pre-treatment trajectories. In high-frequency settings, repeated observations over time can therefore provide informative proxies for hidden confounders, even when these factors are not directly observed. On the other hand, the Lipschitz regularity assumption is required for theoretical justification.

Finally, we propose a framework that learns latent representations from pre-treatment trajectories to capture treatment-relevant proxy information about latent confounders. Our primary objective is accurate individual-level counterfactual estimation, which is then aggregated to estimate the ATT. While the main experiments focus on a common treatment onset time $t_0$ for simplicity and comparability with existing baselines, the framework naturally extends to staggered treatment adoption through treatment-time masking, as detailed in Appendix A.

## 4 Methodology

### 4.1 Overview and Theoretical Intuition

We now formalize the data-generating process underlying the generic potential outcome processes introduced in Section 3.

Let $W$ denote a continuous hidden confounder distributed as $W \sim \mathcal{P}_W$, with density $P(w)$. We assume that the propensity score is described by a function $g$ of $W$ and that the treatment assignment follows a Bernoulli $T \mid W = w \sim \mathrm{Bernoulli}(g(w))$. Conditional on $W = w$, the potential outcome trajectories

$$Y^0(w) = \{Y_t^0(w)\}_{t=1}^L, \qquad Y^1(w) = \{Y_t^1(w)\}_{t=1}^L,$$

follow the conditional distributions $\mathcal{P}_{Y^0|W=w}$ and $\mathcal{P}_{Y^1|W=w}$ respectively. The observed outcome trajectories satisfy:

$$Y = TY^1 + (1 - T)Y^0.$$

We assume that individual units $(Y_i^0, Y_i^1, Y_i, W_i, T_i)_{i \in \mathcal{T} \cup \mathcal{C}}$ are i.i.d. random variables from the joint distribution of $(Y^0, Y^1, Y, W, T)$.

Inspired by the synthetic control framework, for each treated unit $i \in \mathcal{T}$, we assign a set of constant positive weights $B_i = (b_{ij})_{j \in \mathcal{C}}$, forming a weight matrix $B \in \mathbb{R}^{|\mathcal{T}| \times |\mathcal{C}|}$.

These weights are used to estimate the counterfactual outcome under control for treated units. To ensure that the synthetic control provides an unbiased estimate of the untreated outcome for each treated unit, we require that for each $i \in \mathcal{T}$ :

$$\mathbb{E}[Y_{i,t}^0 \mid T = 1] = \mathbb{E}\left[\sum_{j \in \mathcal{C}} b_{ij} Y_{j,t}^0 \mid T = 0\right], \quad \forall t \in \{1, \ldots, L\}. \tag{1}$$

**Proposition 1.** *For any fixed treated unit $i \in \mathcal{T}$ with associated weights $(b_{ij})_{j \in \mathcal{C}}$, equality (1) holds if and only if the following balancing condition is satisfied :*

$$\int \left(\frac{g(w)}{\mathbb{P}(T = 1)} - \sum_{j \in \mathcal{C}} b_{ij} \frac{1 - g(w)}{\mathbb{P}(T = 0)}\right) \mathbb{E}[Y_t^0(w)] P(w) \, dw = 0, \quad \forall t \in \{1, \ldots, L\}. \tag{2}$$

*Proof.* With $g(w) = \mathbb{P}(T = 1 \mid W = w)$, latent ignorability $(Y^0 \perp\!\!\!\perp T | W)$, the law of iterated expectation and Bayes rule give

$$\mathbb{E}(Y_{i,t}^0 \mid T = 1) = \int \frac{g(w)}{\mathbb{P}(T = 1)} \mathbb{E}[Y_{i,t}^0 | W_i = w] P(w) \, dw. \tag{P1}$$

And since units are exchangeable conditional on $W$, we write $\mathbb{E}[Y_{i,t}^0 \mid W_i = w] = \mathbb{E}[Y_t^0(w)]$ which yields:

$$\mathbb{E}[Y_{i,t}^0 \mid T = 1] = \int \frac{g(w)}{\mathbb{P}(T = 1)} \mathbb{E}[Y_t^0(w)] P(w) \, dw.$$

For the synthetic control:

$$\mathbb{E}\left[\sum_{j \in \mathcal{C}} b_{ij} Y_{j,t}^0 \mid T = 0\right] = \int \sum_{j \in \mathcal{C}} b_{ij} \frac{1 - g(w)}{\mathbb{P}(T = 0)} \mathbb{E}[Y_t^0(w)] P(w) \, dw. \tag{P2}$$

Equality of (P1) and (P2) is equivalent to equation 2. Conversely, if equation 2 holds, (P1) = (P2) follows immediately.
□

When treatment is independent of $W$, i.e., $T \perp\!\!\!\perp W$, then $g(w) = \mathbb{P}(T = 1)$ and for $i \in \mathcal{T}$, equation 2 simplifies to:

$$\sum_{j \in \mathcal{C}} b_{ij} = 1,$$

which recovers the classical convex constraint in synthetic control:

$$\hat{Y}_{i,t}^0 = \sum_{j \in \mathcal{C}} b_{ij} Y_{j,t} \quad \text{with } b_{ij} \geq 0, \ \sum_{j \in \mathcal{C}} b_{ij} = 1, \quad \forall t \in \{1, \ldots, L\},$$

where $\hat{Y}_{i,t}^0$ is the estimated potential outcome under control for unit $i$ at time $t$.

However, under hidden confounding, equation (2) reveals that the weights must account for differences in the distribution of $W$ between the treated and control groups. This motivates balancing on the propensity score $g(W)$, which

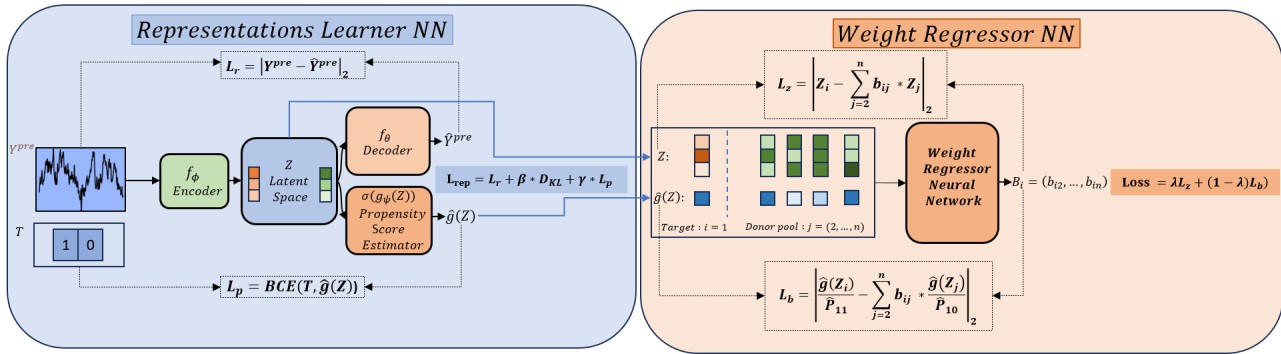

Figure 2: Overview of the proposed algorithm B-Twin. The architecture consists of two main components. The first block is a *representation learning* module based on a two-headed variational autoencoder (VAE). It takes as input the pre-treatment outcomes and produces latent representations $Z$ via an encoder-decoder structure. Simultaneously, it estimates the individual propensity scores $\hat{g}(Z)$ from the latent representations and the treatment indicator. This block is trained using a reconstruction loss $\mathcal{L}_r$, a propensity prediction loss $\mathcal{L}_p$ and a KL divergence term. The second block is a *weight regressor* module. It takes as input the latent representations and estimated propensity scores of a target unit (e.g., a treated individual), along with those of the donor pool (e.g., control units). It outputs a set of balancing weights $B$ that minimize a combined loss function composed of $\mathcal{L}_z$ and $\mathcal{L}_b$, which enforce similarity in both latent space and estimated propensity scores.

summarizes the treatment-relevant information in $W$ (Rosenbaum & Rubin, 1983), leading to the following proxy balancing condition for each unit $i \in \mathcal{T}$:

$$\mathbb{E}[g(W_i) \mid T = 1] = \mathbb{E}\left[\sum_{j \in \mathcal{C}} b_{ij} g(W_j) \mid T = 0\right]. \tag{3}$$

This is a practical relaxation of the stronger balancing condition and focuses on matching the distributions of treatment propensities.

In practice, the confounder $W$ is unobserved, and therefore $g(W)$ cannot be computed directly. To address this, we introduce a learned representation $Z$ obtained from pre-treatment outcome trajectories $Y^{pre}$, which are assumed to be informative of the treatment-relevant latent structure (Assumption 4), so that $Z$ captures information about $W$ relevant for treatment assignment and outcome dynamics. Under this assumption, we estimate a surrogate propensity score $\hat{g}(Z) \approx \mathbb{P}(T = 1 \mid Z)$ using a neural classifier. While $\hat{g}(Z)$ does not recover the true propensity $g(W)$ in general, we show (see Proposition 2) that, under mild regularity conditions given by Assumption 5, it provides an approximation whose error depends on (i) how well pre-treatment outcomes encode the latent confounder, (ii) the reconstruction quality of the learned representation, and (iii) the accuracy of the propensity model.

Our proposed model therefore proceeds as follows: we first learn latent representations $Z$ from pre-treatment trajectories, jointly trained to reconstruct outcomes and predict treatment assignment by estimating $\hat{g}(Z)$. We then optimize weights $B$ to align treated and control groups according to the proxy balancing condition (3). This approach replaces balancing on the unobserved $W$ with a learned, data-driven surrogate that captures treatment-relevant heterogeneity.

## 4.2 Latent Representation Learning via Variational Autoencoding

Our architecture consists of an encoder that maps high-frequency time series data into a low-dimensional latent space $Z$, intended to capture the influence of unobserved confounders $W$. The encoder $f_\phi$ and decoder $f_\theta$ together form a variational autoencoder (VAE) (Kingma et al. 2013; Chung et al. 2015), trained using a reconstruction loss over pre-treatment observations $Y^{pre}$. From the latent representation $Z_i$, $i \in \mathcal{T} \cup \mathcal{C}$, we also estimate the propensity score

$\hat{g}(Z_i)$ via a classifier head. Formally:

$$Z_i = f_\phi(Y_i^{pre}), \quad \hat{Y}_i^{pre} = f_\theta(Z_i), \quad \hat{g}(Z_i) = \sigma(g_\psi(Z_i)),$$

where $\phi, \theta$ and $\psi$ are the parameters of the representation learner, $\sigma$ is the sigmoid activation and $g_\psi$ is a separate multilayer perceptron (MLP) used for propensity estimation.

The latent space $Z$ serves as a proxy for information about $W$ contained in the pre-treatment trajectory, encoding relevant temporal and static features for both outcome modeling and treatment assignment. While our experiments use only pre-treatment time series as input, the framework is flexible: additional covariates $X$ can be concatenated to the input without altering the model's structure. For simplicity and efficiency, we primarily use dense layers, which we found to be well-suited for our high-frequency energy data.

The representation block is trained using the following composite loss function (see Figure 2):

$$\mathcal{L}_{rep} = \mathcal{L}_r + \beta D_{KL} + \gamma \mathcal{L}_p,$$

where

$$\mathcal{L}_r = \|Y^{pre} - \hat{Y}^{pre}\|^2, \quad \mathcal{L}_p = \text{BCE}(T, \hat{g}(Z)), \quad D_{KL} = \text{KL}\left[q(Z \mid Y^{pre}) \,\|\, p(Z)\right].$$

$\beta \in [0, 1]$ and $\gamma \in [0, 1]$ are hyperparameters that control the contribution of the Kullback–Leibler divergence (KL) and Binary Cross Entropy (BCE) for the propensity estimation loss, respectively.

This block is trained on both treated and control units, ensuring that the learned latent space captures variation across the entire population. The following proposition formalizes the connection between the learned propensity score $\hat{g}(Z)$ and the latent propensity score $g(W)$. It shows that the approximation quality depends jointly on the informativeness of the pre-treatment trajectories, the reconstruction quality of the latent representation, and the accuracy of the propensity estimator.

**Proposition 2** (Approximation of the latent propensity). *Under the assumptions above, suppose that: (i) the reconstruction error satisfies*

$$\|Y^{pre} - f_\theta(f_\phi(Y^{pre}))\| \le \varepsilon_Y,$$

*and (ii) the propensity model satisfies*

$$|\hat{g}(Z) - \mathbb{P}(T = 1 \mid Z)| \le \varepsilon_P.$$

*Then,*

$$|\hat{g}(Z) - g(W)| \le \varepsilon_P + 2L_g \varepsilon_W + 2L_g L_h \varepsilon_Y.$$

*Proof. (Proof sketch.)* The proof proceeds in two steps. First, using the latent recoverability assumption together with the Lipschitz continuity of $h$ and $g$, we show that the reconstruction $f_\theta(Z)$ induces an approximation of the latent propensity:

$$|g(W) - g(h(f_\theta(Z)))| \le L_g \varepsilon_W + L_g L_h \varepsilon_Y.$$

This bound combines the recoverability error $\varepsilon_W$ and the reconstruction error $\varepsilon_Y$. Second, we control the discrepancy between the learned propensity $\hat{g}(Z)$ and $g(h(f_\theta(Z)))$. Using the propensity estimation error $\varepsilon_P$, the law of iterated expectations, and the conditional independence assumption $T \perp\!\!\!\perp Z \mid W$, we obtain

$$|\hat{g}(Z) - g(h(f_\theta(Z)))| \le \varepsilon_P + L_g \varepsilon_W + L_g L_h \varepsilon_Y.$$

Combining both inequalities through the triangle inequality yields the final result. The complete proof is provided in Appendix B. □

The proposition shows that, under the outcome informativeness assumption, minimizing both the reconstruction and propensity estimation losses encourages the learned representation to capture the treatment-relevant information contained in the hidden confounder. In particular, when the reconstruction error and propensity prediction error are small, the learned propensity $\hat{g}(Z)$ can be interpreted as an approximation of $g(W)$, with error controlled by recoverability, reconstruction, and propensity-estimation errors. Importantly, the result does not claim exact identification of the hidden confounder itself, but rather capturing the treatment-relevant latent structure required for estimating propensity scores.

### 4.3 Counterfactual Estimation and Weights Learning

Once the variational auto-encoder and propensity score estimator are trained by minimizing the reconstruction loss and propensity estimation loss, we introduce a second neural module to compute balancing weights for counterfactual estimation. This weight regressor takes as input for each target unit $i$ (e.g., treated), its latent representation $Z_i$ and estimated propensity score $\hat{g}(Z_i)$ together with the corresponding quantities $\{Z_j, \hat{g}(Z_j)\}$ for the donor pool (e.g., all control units $j \in \mathcal{C}$). It outputs weights $\{b_{ij}\}$ to construct a synthetic control for unit $i$.

We train this module to minimize a weighted combination of two objectives:

$$\mathcal{L}_{\text{match}} = \lambda \mathcal{L}_z + (1 - \lambda)\mathcal{L}_b,$$

where $\lambda \in [0, 1]$ balances the contribution of the latent and propensity terms. The losses are defined as :

$$\mathcal{L}_z = \frac{1}{|\mathcal{T}|} \sum_{i \in \mathcal{T}} \left\| Z_i - \sum_{j \in \mathcal{C}} b_{ij} Z_j \right\|^2, \qquad \mathcal{L}_b = \frac{1}{|\mathcal{T}|} \sum_{i \in \mathcal{T}} \left\| \frac{\hat{g}(Z_i)}{\hat{P}_{11}} - \sum_{j \in \mathcal{C}} b_{ij} \frac{\hat{g}(Z_j)}{\hat{P}_{10}} \right\|^2,$$

with :

$$\hat{P}_{11} = \frac{1}{|\mathcal{T}|} \sum_{i \in \mathcal{T}} \hat{g}(Z_i), \qquad \hat{P}_{10} = \frac{1}{|\mathcal{C}|} \sum_{i \in \mathcal{C}} \hat{g}(Z_i), \tag{4}$$

where $\mathcal{L}_b$ loss is derived from Equation (3). Normalization by $\hat{P}_{11}$ and $\hat{P}_{10}$ is added to avoid instability due to scale differences between treated and control groups, leading to degenerate solutions when treatment probabilities are skewed.

### 4.4 Training Procedure and Hyperparameters tuning

Our model is trained in two stages. First, the representation block learns latent embeddings by minimizing a combined loss of reconstruction error, KL divergence ($\beta = 0.005$), and propensity prediction loss ($\gamma = 0.1$). We use the Adam optimizer (learning rate $10^{-3}$), dropout rate of 0.3, and gradient clipping at 1.0.

After freezing the representations, the weight regressor is trained to learn individual-specific balancing weights using a latent matching loss ($\lambda = 0.7$). This phase also uses Adam ($10^{-4}$) with the same regularization settings. Importantly, although matching requires the full set of potential matches, the query group can be processed in mini-batches while keeping the full donor pool. This strategy allows the model to scale efficiently to large datasets, enabling inference on high-dimensional, individual-level data without prohibitive memory requirements.

Hyperparameters were chosen for empirical stability, convergence and decreasing loss but could be tuned automatically via cross-validation or using a placebo dataset.

### 4.5 Counterfactual Estimation

After training the weight regressor, we obtain estimated matching weights $\hat{B} = (\hat{b}_{ij})_{i \in \mathcal{T}, j \in \mathcal{C}}$. Using the learned latent representations and these estimated weights, the counterfactual outcome for a treated unit $i \in \mathcal{T}$ at time $t$ is estimated as:

$$\hat{Y}_{i,t}^0 = \sum_{j \in \mathcal{C}} \hat{b}_{ij} Y_{j,t}.$$

The corresponding individual treatment effect (ITE) is then given by:

$$\hat{\text{ITE}}_{i,t} = Y_{i,t} - \hat{Y}_{i,t}^0, \quad t \geq t_0.$$

### 4.6 Average Treatment Effect on the Treated (ATT)

Using importance weighting (Kloek & Van Dijk, 1978) with the latent propensity score $g(W)$, the ATT can be rewritten as:

$$\text{ATT} = \mathbb{E}\left[\frac{1}{L - t_0 + 1}\sum_{t=t_0}^{L}\left(Y_t^1 - Y_t^0\right) \,\bigg|\, T = 1\right] = \mathbb{E}_W\left[\frac{g(W)}{\mathbb{P}(T=1)}\mathbb{E}\left[\frac{1}{L - t_0 + 1}\sum_{t=t_0}^{L}\left(Y_t^1 - Y_t^0\right) \,\bigg|\, W\right]\right]. \quad (5)$$

**Estimation**

A standard empirical estimator of the ATT based on treated units is:

$$\hat{\text{ATT}} = \frac{1}{L - t_0 + 1}\sum_{t=t_0}^{L}\frac{1}{|\mathcal{T}|}\sum_{i\in\mathcal{T}}(Y_{i,t} - \sum_{j\in\mathcal{C}}\hat{b}_{ij}Y_{j,t}). \quad (6)$$

Motivated by Equation (5), we instead propose an importance-weighted estimator that leverages samples from the full population. Equation (5) rewrites the ATT as an expectation over the marginal latent distribution $\mathcal{P}_W$ using importance weighting. Consequently, the corresponding empirical estimator is no longer restricted to treated units only, but can leverage samples from the entire population ($\mathcal{T} \cup \mathcal{C}$), weighted according to their estimated treatment propensity. To symmetrize the estimator, we additionally compute reverse matching weights $(\hat{b}_{ji})_{j\in\mathcal{C}, i\in\mathcal{T}}$ from control units to treated units, allowing counterfactual outcomes to also be constructed for control samples. This leads to the following symmetrized empirical estimator, where both treated and control units contribute to the empirical approximation of the expectation over $W$. Given latent representations $Z$, estimated propensity scores $\hat{g}(Z)$, observed outcomes $Y$, and estimated matching weights, we approximate ATT as:

$$\hat{\text{ATT}} = \frac{1}{n(L - t_0 + 1)}\sum_{t=t_0}^{L}\left[\sum_{i\in\mathcal{T}}\left(\frac{\hat{g}(Z_i)}{\hat{P}_{11}}Y_{i,t} - \sum_{j\in\mathcal{C}}\hat{b}_{ij}\frac{\hat{g}(Z_j)}{\hat{P}_{10}}Y_{j,t}\right) + \sum_{j\in\mathcal{C}}\left(\sum_{i\in\mathcal{T}}\hat{b}_{ji}\frac{\hat{g}(Z_i)}{\hat{P}_{11}}Y_{i,t} - \frac{\hat{g}(Z_j)}{\hat{P}_{10}}Y_{j,t}\right)\right]. \quad (7)$$

Here : $n = |\mathcal{T} \cup \mathcal{C}|$ and $\hat{P}_{11}$ and $\hat{P}_{10}$ are defined in equation (4).

In our ablation study, we empirically observe that this approach yields more robust and stable estimates of ATT than the standard estimator in Eq. (6), particularly when propensity score estimation is noisy.

## 5 Experiments

We evaluate our method across different settings: simulated datasets, a semi-synthetic version of the mimic dataset, a real-world electricity behavioral challenge and its semi-synthetic version for comparison. These experiments help assess robustness, bias, and scalability, comparing B-Twin to classical econometric models and neural baselines in terms of accuracy and efficiency.

### 5.1 Simulated data, ablation and tuning analysis

In this section, we illustrate our approach on two simulation models, and conduct an ablation study and hyperparameters analysis for the simplest case.

#### 5.1.1 Toy Dataset 1

In the first simulation model, a hidden confounder influences both the treatment assignment and the outcome, with its effect on the outcome increasing gradually over time, in order to break the parallel trends assumption. This design

makes the confounding effect difficult to infer from the pre-treatment observations, posing a challenge for methods that assume parallel trends or rely on short-term observation.

Untreated outcomes for each individual $i$ at time $t$ are generated using the following process:

$$Y_{i,t}^0 = 0.5\, q_t\, W_i^2 + \alpha\, t\, \mathbf{1}_{W_i > 0.5} + \epsilon_{it}, \quad \forall i, t,$$

where $q_t$ is a time-varying factor following an AR(1) process: $q_t = \rho\, q_{t-1} + \epsilon_t$, and $W_i \sim \mathcal{U}[0,1]$ is an unobserved confounder. The noise terms follow $\epsilon_t \sim \mathcal{N}(0, \sigma)$. This formulation is inspired by the latent factor model in SyncTwin (Qian et al., 2021).

Treatment begins at time $t = t_0$, with treatment assignment governed by the following logistic function:

$$\mathbb{P}(T_i = 1 \mid W_i = w) = g(w) = \frac{1}{1 + \exp(-5(w - 0.5))}, \tag{8}$$

introducing selection bias between treated and control units. The treated outcome is defined as:

$$Y_{it}^1 = Y_{it}^0 + \tau_i\, \mathbf{1}\{T_i = 1,\, t \geq t_0\}.$$

where $\tau_i$ is the individual treatment effect.

**Simulation Parameters**

| Setting | $\sigma$ | $\alpha$ | $\tau_i$ | $g(w)$ |
|---------|----------|----------|----------|--------|
| a | 5 | 0.05 | 1.54 | 0.5 |
| b | 1 | 0 | 1.54 | logistic (8) |
| c | 5 | 0.05 | 1.54 | logistic (8) |
| d | 5 | 0.05 | $4\log(1 + W_i)$ | logistic (8) |

Table 1: Simulation settings: outcome noise ($\sigma$), confounding strength ($\alpha$), treatment effect ($\tau_i$), and treatment assignment function $g(w)$.

We evaluate four configurations summarized in Table 1, varying noise ($\sigma$), confounding strength ($\alpha$), treatment effect ($\tau_i$), and propensity score $g(w)$. The first setting represents a randomized scenario, where standard methods perform well. The second setting has minimal confounding and noise. The third is our primary benchmark with strong noise and a constant effect. The fourth introduces treatment effect heterogeneity via dependence on the hidden confounder $W$. The dataset contains 500 individuals with outcomes of length 168 each, and treatment starts at $t_0 = 84$. Neural methods estimate ATT by averaging individual effects, while classical baselines use group-level averages (a single aggregated curve for the treated group) and generate a single counterfactual. Although this comparison favors traditional methods computationally, we adopt it to reduce runtime. Despite this, our approach remains competitive and gives consistent and stable results.

**Evaluation Strategy and Baselines**

We evaluate all methods based on their ability to recover the average treatment effect on the treated (ATT) over 100 Monte Carlo replications. Results are summarized using boxplots, allowing us to assess both accuracy and stability. A method is considered reliable if its estimates are consistently close to the ground truth and exhibit low variance across runs.

Our baselines include classical econometric approaches such as Ordinary Least Squares (OLS) and Difference-in-Differences (DiD), which rely on linearity and the parallel trends assumption. We also consider several variants of Synthetic Control (SC): the standard formulation, regularized versions using lasso, ridge, and elastic-net penalties, and a dynamic time warping (DTW) variant designed to improve temporal alignment. In addition, we include Synthetic

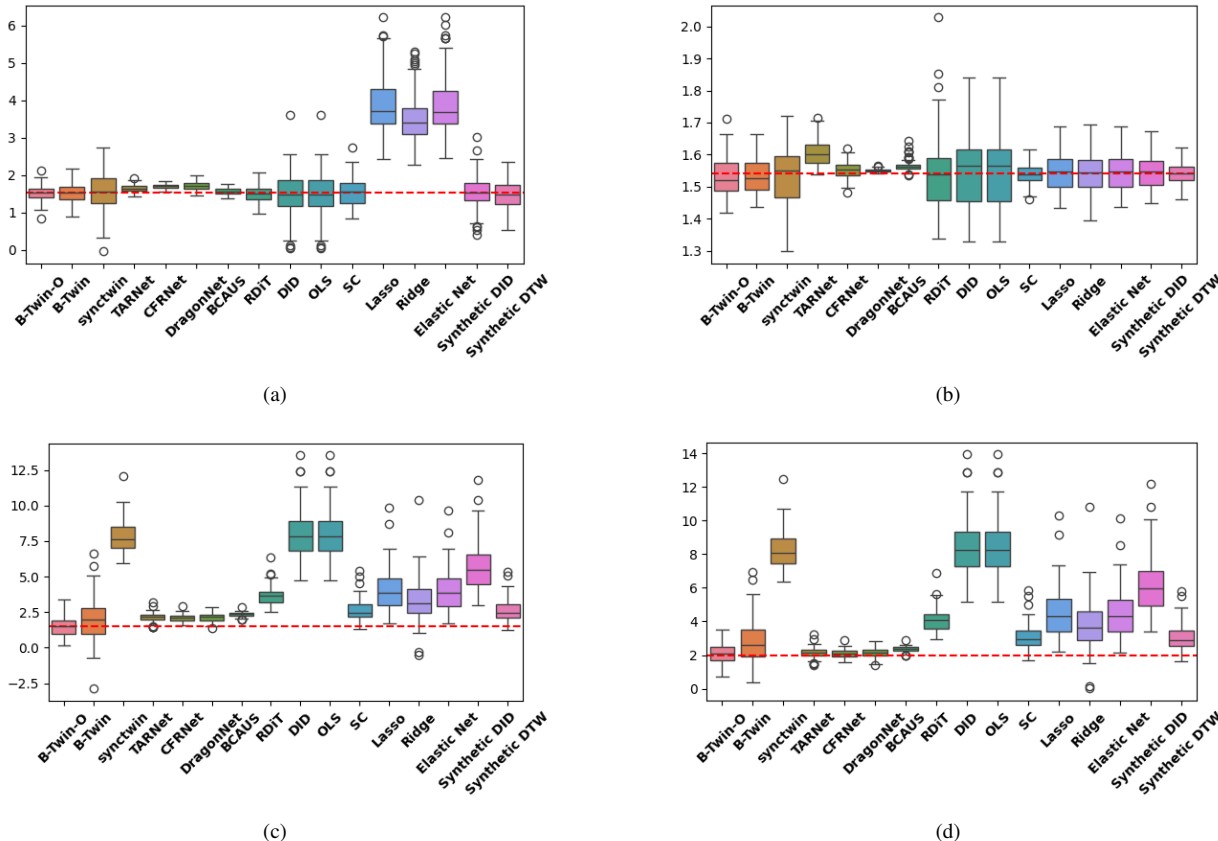

Figure 3: ATT estimation across four settings (see Table 1), 100 Monte Carlo simulations. (a) Randomized experiment; (b) low noise and low confounding; (c) high noise and strong confounding; (d) heterogeneous treatment effects. The red dashed line denotes the ground-truth ATT.

Difference-in-Differences (SDID), which combines elements of DiD and SC to balance units and time periods. We additionally include Regression Discontinuity in Time (RDiT) (Hausman & Rapson, 2018), which estimates treatment effects by treating intervention time as a temporal cutoff and measuring local discontinuities around treatment onset.

Among neural network baselines, we include SyncTwin, a representation-learning method that uses temporal autoencoding to model latent confounding while retaining a synthetic-control-style parametric estimation. We also evaluate state-of-the-art neural causal inference methods designed for cross-sectional settings, including TARNet, CFRNet, DragonNet, and BCAUSS. These models estimate potential outcomes via learned representations and outcome regressors, and we include them to examine how such approaches perform when applied to high-frequency time series data with hidden confounding.

Together, these baselines span a wide range of econometric, synthetic control, and learning-based approaches. Our proposed method, **B-Twin**, is compared against all baselines, and we additionally report an oracle variant, **B-Twin-O**, which uses ground-truth propensity scores to isolate the effect of propensity estimation.

## Results

Figure 3 summarizes ATT estimation performance across four experimental settings. In the randomized setting (a), where treatment assignment is independent of the hidden confounder, most methods achieve low bias and variance, as expected.

In the low-noise, weak-confounding setting (b), all the methods estimate the true effect, with some variability across methods.

Under high noise and strong confounding (c), classical approaches degrade substantially, and SyncTwin struggles due to its reliance on temporal reconstruction via recurrent architectures. Interestingly, neural baselines such as TARNet, CFRNet, DragonNet, and BCAUSS achieve strong performance in terms of accuracy and confidence interval width, despite not being explicitly designed for this setting. We hypothesize that, by treating pre-treatment outcomes as covariates, these models implicitly capture predictive relationships in the data-generating process and extrapolate potential outcomes through outcome regression.

Across all settings, B-Twin remains stable, closely tracking the oracle B-Twin-O, which highlights its robustness to both noise and hidden confounding.

In the heterogeneous treatment effect scenario (d), B-Twin-O and B-Twin again recover accurate ATT estimates, while classical baselines and SyncTwin exhibit substantial bias. Although neural baselines continue to perform well in this scenario, their reliance on outcome regression may make them more sensitive to changes in the data-generating process. As we demonstrate in the next set of simulations, their performance may deteriorate when non-stationarity causes the relationship between pre- and post-treatment outcomes to shift over time.

### 5.1.2  Toy Dataset 2

In this experiment, we modify the data-generating process (DGP) such that pre-treatment and post-treatment outcomes are generated by different mechanisms, independently of the treatment itself, while remaining influenced by the hidden confounder.

The outcome is generated using the following generative process :

$$Y_{i,t}^0 = 0.5\, q_t W_i^2 + \alpha t\, \mathbf{1}_{\{W_i > 0.5\}} + \epsilon_{i,t}, \quad \forall i,\ t \le t_0,$$

$$Y_{i,t}^0 = 0.5\, q_t W_i^2 + \alpha t\, \mathbf{1}_{\{W_i > 0.5\}} + \beta_1 \sin(kt) W_i + \beta_2 q_t^2 W_i + \epsilon_{i,t}, \quad \forall i,\ t > t_0.$$

where $q_t$, $W_i$ and $\epsilon$ are chosen as before. This setting introduces non-stationarity between the pre- and post-intervention periods. We further increase the dataset size to 10,000 individuals to assess how different methods scale under these conditions. In the following experiments, we chose $\beta_1 = 3$ , $\beta_2 = 0.2$ and $k = 0.1$.

### Results

Figure 4 reports the distribution of ATT estimates across 100 Monte Carlo runs for the four experimental settings described in Table 1. In the randomized treatment assignment setting (a), all methods exhibit low bias and variance, as expected in the absence of confounding.

In setting (b), where observational noise remains low and the parallel trends assumption approximately holds, the methods approximately estimate a value that is close to the true treatment effect. However, performances degraded compared to the first simulation, where the data-generating process (DGP) was stationary. In settings (c) and (d), neural outcome-regression baselines such as TARNet, CFRNet, DragonNet, and BCAUSS show a marked increase in bias and variance compared to the previous cases. This degradation highlights their reliance on learning a stable data-generating process (DGP) from pre-treatment outcomes and extrapolating it to the post-treatment period, an assumption that is weakened in this setting.

Across all scenarios, B-Twin consistently tracks its oracle counterpart (B-Twin-O), demonstrating robustness to hidden confounding and noise while maintaining low variance, even in the more challenging regimes (c) and (d) where confounding is strong and the DGP evolves over time. These results suggest that matching individuals in a learned latent space, rather than directly generating potential outcomes via outcome regression, seems more robust under non-stationarity, an important property for counterfactual estimation in high-frequency observational time series.

Finally, we note that classical approaches such as synthetic control, are applied here to a single aggregated treated trajectory. While this simplifies the estimation task relative to individual-level counterfactual generation, it is included for reference, but we were unable to run it on individual level. In practice, classical synthetic control methods are computationally infeasible and numerically unstable at the individual level for large-scale, high-frequency datasets (see training times in Figure 7, 8 and 9), whereas B-Twin is explicitly designed to operate at this granularity.

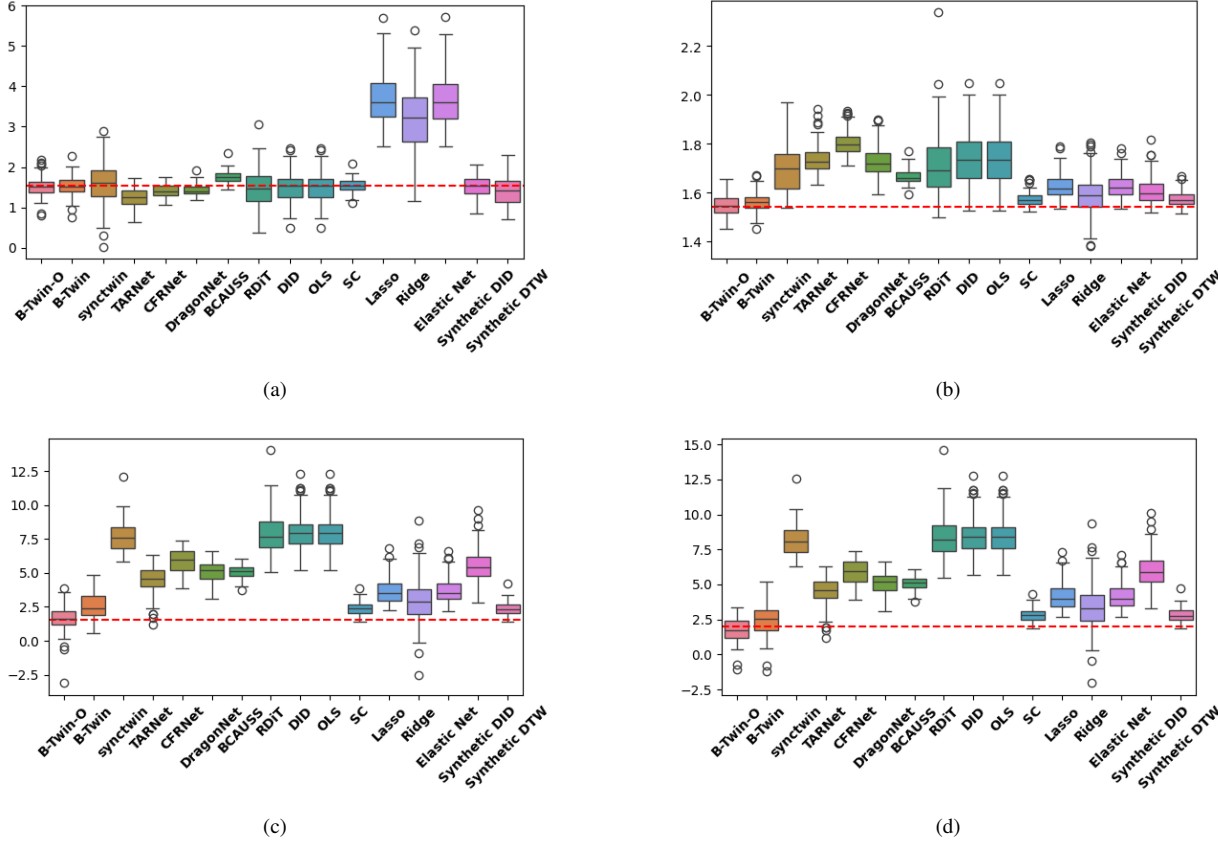

Figure 4: ATT estimation across four settings (see Table 1), 100 Monte Carlo simulations. (a) Randomized experiment; (b) low noise and low confounding; (c) high noise and strong confounding; (d) heterogeneous treatment effects. The red dashed line denotes the ground-truth ATT.

### 5.1.3 Ablation Study

We perform an ablation study on the first simulated dataset, which provides a clear and controlled setting for analyzing architectural variants and hyperparameter sensitivity.

### Architecture Variants

We first analyze several architectural variants of B-Twin to evaluate the impact of key modeling decisions. As summarized in Table 2, we examine (i) whether propensity scores are estimated or known, (ii) the role of different loss components, and (iii) whether the Average Treatment Effect on the Treated (ATT) is computed using only treated units or the full population.

**Baselines without Propensity Balancing.** **B-Twin-C1** learns latent representations solely from outcome trajectories $Y$ using the reconstruction loss $L_r$. Counterfactuals are constructed via convex combinations of control units, similarly to the SyncTwin framework, but using dense layers instead of recurrent architectures.

**B-Twin-C2** extends **B-Twin-C1** by adding a propensity estimation head trained jointly with the reconstruction loss $(L_r + \beta L_p)$, while still enforcing convex matching weights. This variant isolates the effect of learning propensity scores without relaxing the convexity constraint.

**Oracle Variants.** To assess the upper bound of performance, we consider oracle variants with access to the true propensity scores:

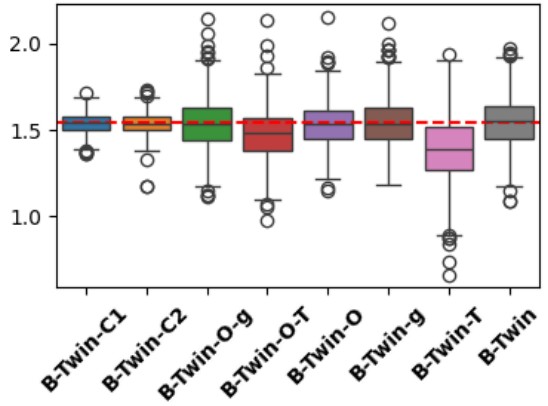 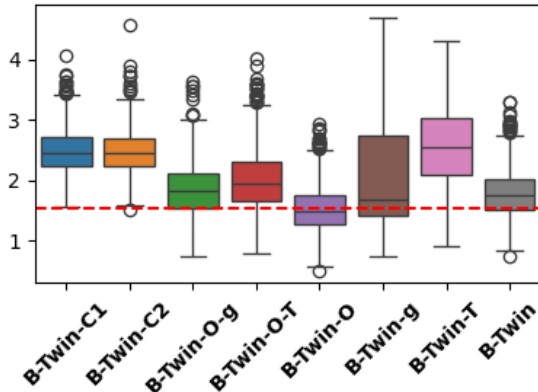

Figure 5: Ablation study results. Left: randomized setting; Right: high-noise, confounded setting. The red dashed line denotes the true ATT.

| Variant | Representation Learner | | Weight Regressor | ATT |
|---|---|---|---|---|
| | Inputs | Propensity Supervision | Inputs | Population |
| B-Twin-C1 | $Y$ | – | $Z$ | Treated |
| B-Twin-C2 | $Y$ | T | $Z$ | Treated |
| B-Twin-O-g | $Y$ | $g$ | $Z, g$ | Treated |
| B-Twin-O-T | $Y$ | T | $Z, g$ | Treated |
| B-Twin-O | $Y$ | T | $Z, g$ | Treated + Control |
| B-Twin-g | $Y$ | $g$ | $Z, \hat{g}$ | Treated |
| B-Twin-T | $Y$ | T | $Z, \hat{g}$ | Treated |
| B-Twin | $Y$ | T | $Z, \hat{g}$ | Treated + Control |

Table 2: Ablation study variants. For each configuration, We report the inputs to the representation learning block and the weight regression block, the use of true or estimated propensity scores, and the population used to compute the ATT.

- **B-Twin-O-g**: Uses the true propensity score $g$ for both supervision and weights regression.

- **B-Twin-O-T**: Supervises the representation using treatment assignment $T$ while the true $g$ is used in the weight regressor; ATT is computed using treated units only.

- **B-Twin-O**: is similar to **B-Twin-O-T** except that ATT is estimated using contributions from both treated and control units.

**Proposed Method.** **B-Twin** is our full proposed model, which relies on estimated propensity scores $\hat{g}$:

- **B-Twin-g**: Trains the representation block using the true $g$ but performs matching using estimated propensities $\hat{g}$.

- **B-Twin-T**: Learns both latent representations and $\hat{g}$ from outcomes $Y$ and treatment $T$, and computes ATT using treated units only.

- **B-Twin**: is similar to **B-Twin-T** except that ATT is estimated using contributions from both treated and control units.

Results in Figure 5 demonstrate the advantages of the full B-Twin model. As expected, the oracle variants, particularly **B-Twin-O**, achieve the most accurate and stable estimates, with confidence intervals tightly centered around the true ATT, providing an upper bound on achievable performance and supporting the proposed balancing formulation. Importantly, the full **B-Twin** model closely matches this oracle performance despite relying solely on estimated propensity scores, indicating that accurate treatment effect recovery does not require access to true propensities.

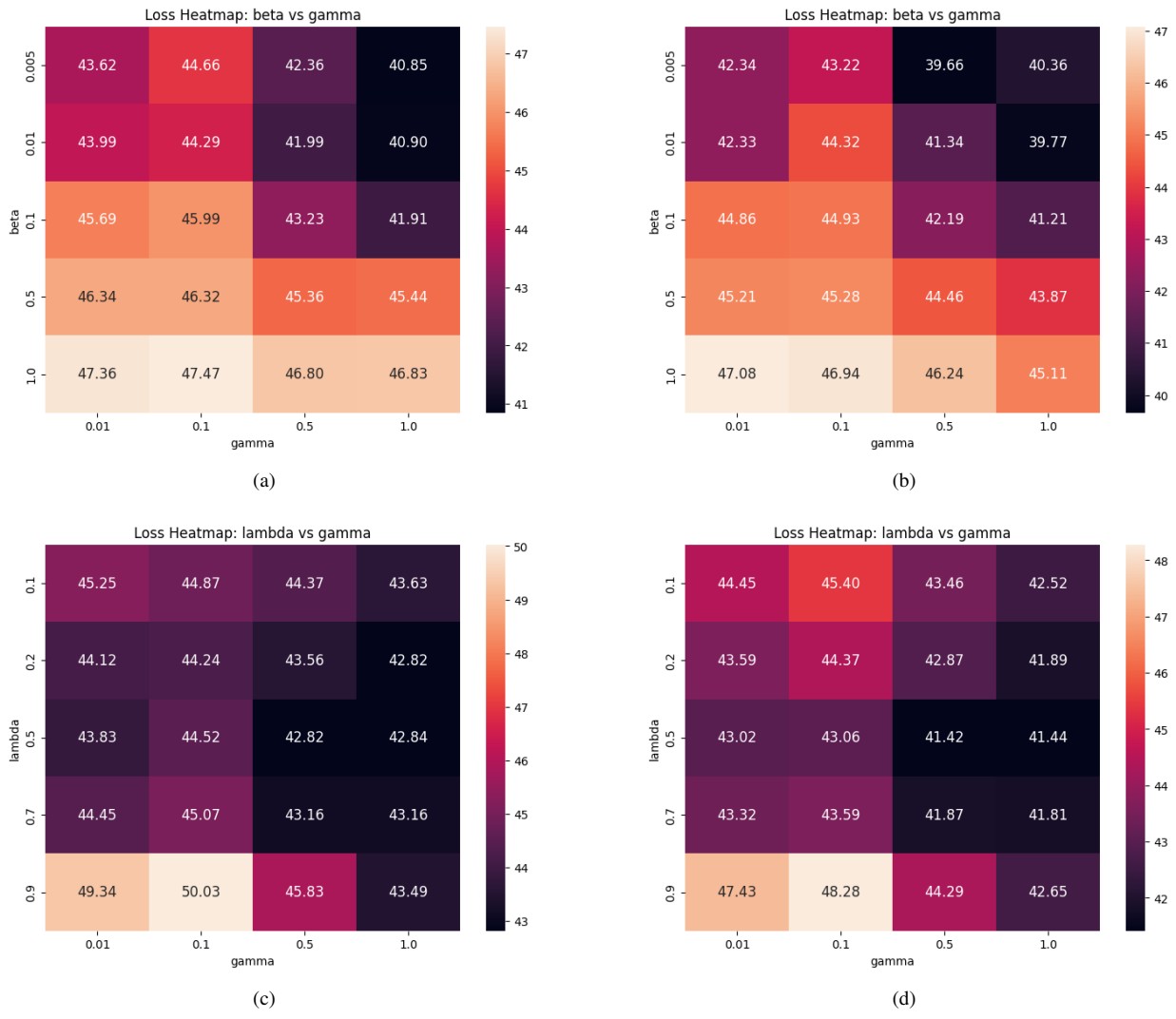

Figure 6: Results on the simulated dataset for simulation 1, cases (c) and (d) in table 1, showing the sensitivity of ITE error to hyperparameters $\beta$, $\gamma$, and $\lambda$.

Examining **B-Twin-C1** and **B-Twin-C2**, which can be viewed as dense-layer analogues of SyncTwin, we observe improved estimation stability relative to SyncTwin (see figure 3 (c)), suggesting that dense architectures are better suited for high-frequency time series than recurrent models in this setting. However, their inferior performance compared to B-Twin highlights the importance of replacing classical convexity constraints with explicit propensity-based balancing. Overall, these results confirm that the combination of learned representations and propensity-driven balancing provides better treatment effect estimation.

## Hyperparameters analysis

We further examine the sensitivity of B-Twin to the hyperparameters $\beta$, $\gamma$, and $\lambda$ on the same simulated dataset, which includes hidden confounding under both constant and heterogeneous treatment effects. Sensitivity is evaluated using the mean error of individual treatment effect (ITE) estimates, which serves as a proxy for ATT error since our estimator aggregates individual counterfactuals.

The resulting heatmaps in figure 6 indicate that performance is largely stable across a wide range of hyperparameter values. In particular, the differences between optimal and suboptimal configurations are marginal, suggesting limited

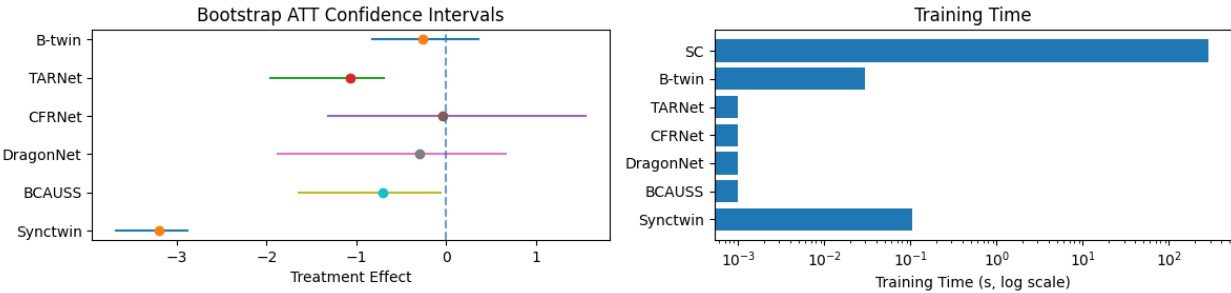

Figure 7: ATT estimation on the semi-synthetic MIMIC-III dataset.

sensitivity to precise tuning. Notably, although the selected hyperparameters ($\beta = 0.005$, $\gamma = 0.1$, $\lambda = 0.7$) are not globally optimal, they still yield strong performance across both simulation settings.

Examining the interaction between $\lambda$ and $\gamma$, we observe that estimation error remains relatively stable over broad regions of the parameter space, further indicating robustness to hyperparameter choice. Overall, these results suggest that selecting hyperparameters to ensure stable convergence and decreasing loss is sufficient to obtain reliable performance, as this promotes the construction of a well-behaved latent space.

Finally, we note that hyperparameter selection via placebo experiments is feasible in practice. In preliminary experiments, we observed that hyperparameters chosen to minimize estimated treatment effects on placebo data also led to low estimation error on real datasets. While we do not report these results in detail, this suggests a practical tuning strategy for applied settings.

## 5.2 Real Data

Across all real-data experiments presented in the following sections, we restrict the comparison to neural methods, including **B-Twin**, **SyncTwin**, **TARNet**, **CFRNet**, **DragonNet**, and **BCAUSS**. These approaches are the most relevant to our setting, as they operate at the individual level and are designed to scale to large, high-dimensional time series data. Classical methods such as Synthetic Control (SC) are therefore excluded from direct comparison, as individual-level estimation is computationally infeasible in these settings. SC runtimes are nevertheless reported for reference. Figures report bootstrap confidence intervals for the ATT (left panel) computed using $n = 100$ resamples, and training times normalized by dataset size on a logarithmic scale (right panel).

### 5.2.1 MIMIC-III Semi-Synthetic Study

We evaluate our approach on a semi-synthetic dataset constructed from the MIMIC-III (Johnson et al., 2016) database, focusing on patients admitted to the intensive care unit (ICU). The outcome of interest is the mean arterial pressure (MAP), and the treatment corresponds to the administration of vasopressors.

Clinical measurements in MIMIC-III are irregularly sampled, and patients exhibit heterogeneous lengths of stay. To obtain comparable time series, we restrict the analysis to patients with an ICU stay of at least 72 hours and interpolate missing measurements to produce regularly sampled trajectories. We retain the first 72 hours of observations following ICU admission for each patient.

To construct a controlled yet realistic evaluation setting with known ground truth, we generate a semi-synthetic treatment assignment in a placebo framework. Specifically, treatment probabilities are induced by applying a logistic function of patient age, introducing selection bias between treated and untreated individuals while preserving the original outcome trajectories. Treatment is assumed to begin at time $t_0 = 60$. Only patients who did not receive vasopressors during the observation window are included, ensuring that the true treatment effect is zero.

Figure 7 reports bootstrap confidence intervals for the ATT on the semi-synthetic MIMIC-III dataset, where the true treatment effect is zero (indicated by the dashed vertical line).

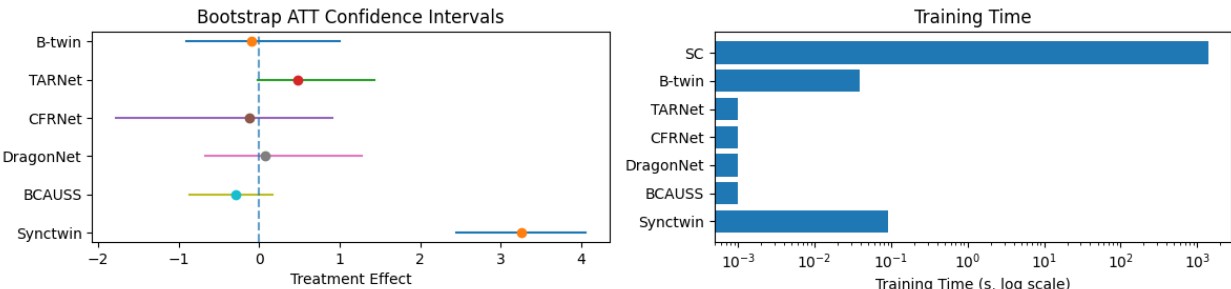

Figure 8: ATT estimation on the semi-synthetic Sowee dataset.

The methods exhibit markedly different behaviors. B-Twin produces a confidence interval that overlaps zero and remains relatively narrow compared to several neural baselines, indicating stable ATT estimates under bootstrap resampling. TARNet and BCAUSS yield confidence intervals that are shifted away from zero, suggesting biased ATT estimates in this setting. CFRNet and DragonNet exhibit wider intervals that overlap zero, reflecting higher estimation variability.

SyncTwin shows a strongly negative ATT estimate with a confidence interval that does not overlap zero, indicating substantial bias under the induced selection mechanism.

The right panel reports training times on a logarithmic scale. Synthetic Control is orders of magnitude more computationally expensive and is included only for reference. All neural methods scale efficiently, with TARNet, CFRNet, DragonNet, and BCAUSS being the fastest due to their outcome-regression structure. B-Twin incurs a moderate additional computational cost relative to these baselines, reflecting the explicit matching and balancing steps, while remaining substantially faster than both SyncTwin and Synthetic Control.

Overall, the results highlight a trade-off between computational efficiency and estimation behavior. B-Twin achieves ATT estimates that overlap the ground truth while maintaining reasonable uncertainty and scalability, suggesting that combining learned representations with explicit balancing can improve stability in this setting.

### 5.2.2 Real-World Electricity Challenge Dataset

In this section, we evaluate our method on a real-world electricity consumption dataset provided by Sowee, a subsidiary of EDF. The data were collected during a behavioral energy-saving challenge conducted between 2021 and 2022. In this challenge, participating households were offered a €50 incentive if they reduced their electricity consumption by at least 5% between two consecutive winter periods. The dataset contains 7,924 individuals with 302 days of daily electricity consumption data spanning both pre- and post-intervention periods (November 2021–March 2022 and November 2022–March 2023). The treatment corresponds to voluntary participation in the energy-saving challenge, while the outcome of interest is daily electricity consumption. To detect selection bias in the treatment adoption, we rely exclusively on pre-treatment consumption trajectories which serve as proxies for behavioral confounding factors such as habits, routines, and baseline energy usage. We first evaluate the methods on a semi-synthetic version of this dataset with a known ground-truth treatment effect. We then report results on the full real dataset for illustration.

**Semi-Synthetic Placebo Dataset (True Effect = 0)**

Since the real dataset does not provide ground-truth treatment effects, we construct a semi-synthetic placebo dataset based on the Sowee data. Treatment assignment is generated artificially to induce selection bias while preserving the original outcome trajectories, ensuring that the true treatment effect remains zero. This setting allows us to assess whether methods spuriously detect treatment effects in the presence of induced selection bias.

Figure 8 reports bootstrap confidence intervals for the ATT on the semi-synthetic Sowee dataset, where the true treatment effect is zero (indicated by the dashed vertical line).

The left panel shows substantial differences across methods. B-Twin yields an ATT estimate close to zero with a confidence interval that overlaps the null effect, indicating stable estimation under bootstrap resampling. CFRNet

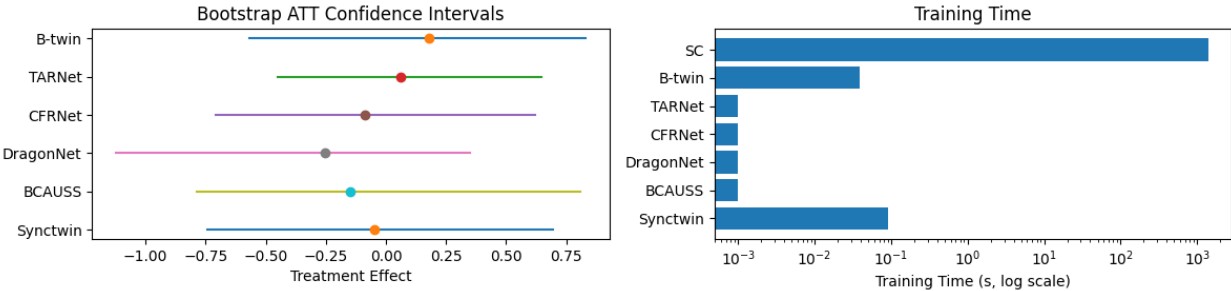

Figure 9: ATT estimation on the full Sowee dataset. Treatment effect is unknown.

and DragonNet also produce estimates centered near zero, though with wider confidence intervals, reflecting higher variability. In contrast, TARNet exhibits a positive bias, with its confidence interval shifted away from zero. BCAUSS produces a slightly negative estimate with an interval that partially overlaps zero.

SyncTwin exhibits a large positive ATT estimate and a confidence interval that does not overlap the null effect, indicating a strong spurious effect in this placebo setting.

The right panel reports training times on a logarithmic scale. Synthetic Control is several orders of magnitude more computationally expensive. Outcome-regression neural models (TARNet, CFRNet, DragonNet, and BCAUSS) are the fastest. B-Twin incurs a higher computational cost than these baselines due to its explicit matching and balancing steps, but remains substantially faster than Synthetic Control.

Overall, this placebo experiment highlights differences in robustness to induced selection bias. While several neural baselines recover ATT estimates near zero, their uncertainty varies substantially. B-Twin consistently produces estimates centered on the null effect while maintaining explicit matching-based counterfactual construction.

### The Real Dataset

We now estimate the ATT on the original Sowee dataset. Since the ground truth is unknown, the following results should be interpreted as an illustration of method behavior on real observational data rather than as validation of the true treatment effect. Agreement across methods and consistency with expected behavior provide qualitative checks only.

Figure 9 reports bootstrap confidence intervals for the ATT on the full Sowee dataset, where the true treatment effect is unknown. The left panel shows that all methods produce ATT estimates concentrated around small values, with confidence intervals largely overlapping the range between approximately $-1$ and $0.75$. This indicates a broad agreement across methods that the average effect of the intervention is limited in magnitude.

The right panel reports training times on a logarithmic scale. As before, neural outcome-regression models are the fastest, while synthetic control is several orders of magnitude slower and computationally impractical at the individual level for a dataset of this size. B-Twin occupies an intermediate position: it remains orders of magnitude faster than synthetic control while retaining an explicit matching-based estimation strategy that constructs counterfactuals from observed data rather than relying on outcome extrapolation.

Overall, these results suggest broad agreement across methods that the average effect of the intervention is small. B-Twin achieves comparable estimates while remaining interpretable and more scalable than individual-level synthetic-control, making it practical for large real-world datasets.

## 6 Limitations

While B-Twin provides a flexible and interpretable framework for counterfactual estimation under hidden confounding, its validity relies on the latent recoverability assumption introduced in Section 3. In particular, B-Twin assumes that the treatment-relevant hidden confounding structure can be approximately recovered from pre-treatment trajectories. If the hidden confounders do not leave a sufficiently informative signature in the observed history, then the

learned representation may fail to preserve the treatment-relevant information required for balancing, and the learned propensity score $\hat{g}(Z)$ may provide a poor approximation of the latent propensity $g(W)$. Consequently, B-Twin should not be interpreted as identifying causal effects under arbitrary hidden confounding, but rather under hidden confounding that is partially recoverable from observed pre-treatment dynamics. In practice, the quality of the method depends jointly on reconstruction accuracy and propensity estimation accuracy. Poor reconstruction quality, misspecified latent representations, or inaccurate propensity estimation may lead to imperfect balancing and biased counterfactual estimates. In addition, very small propensity score values (i.e., limited overlap between treated and control populations) may introduce instability in both training and ATT estimation due to the sensitivity of importance weighting. Finally, B-Twin introduces additional computational overhead compared to standard outcome-regression methods because of the explicit matching step. As a result, in settings where the data-generating process is approximately stationary, extrapolation is reliable, and interpretability is not required, simpler outcome-regression approaches may be preferable.

## 7 Conclusion

We proposed **B-Twin**, a neural framework for estimating treatment effects from time series data in the presence of hidden confounding. B-Twin combines representation learning with individual-level propensity score estimation to construct interpretable, synthetic-control-style counterfactuals based on explicit matching rather than outcome extrapolation.

Across synthetic, semi-synthetic, and real-world experiments, B-Twin exhibits stable performance under hidden confounding, noise, and non-stationary data-generating processes. Its robustness stems from anchoring counterfactual estimation to observed control trajectories, which mitigates the sensitivity of outcome-regression-based approaches when temporal dynamics shift between pre- and post-treatment periods.

By decoupling representation learning from matching, B-Twin provides a principled alternative to outcome-regression approaches for counterfactual estimation in time series. Its strengths are most pronounced in settings where hidden confounding and non-stationary dynamics limit the reliability of extrapolation-based methods, and where interpretability through explicit weighting is desirable.

Future work will explore improved representation learning for capturing complex latent structure and extend the framework to model heterogeneous and time-varying treatment effects more explicitly across individuals.

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

# A    Handling Staggered Treatment Adoption

**Treatment process.**    In the staggered adoption setting, treatment may start at different times $t_i$ across individuals. We define the treatment process

$$A_{i,t} = \mathbf{1}\{t \geq t_i\},$$

where $t_i$ denotes the treatment onset time for individual $i$. For untreated individuals, pseudo-treatment times are sampled randomly from observed treatment-times to construct comparable pre-treatment windows. This allows pre-treatment windows to be defined consistently across treated and control units in the staggered setting.

**Pre-treatment encoding.**    The encoder receives both the outcome trajectory $Y_{i,t}$ and the treatment process $A_{i,t}$. To ensure that the learned representation only captures pre-treatment information, the encoder operates on the masked trajectory

$$Y_{i,t}(1 - A_{i,t}),$$

which removes post-treatment observations from the representation learning stage while preserving the temporal structure of the sequence.

**Latent representation and matching.**    Using the masked outcome trajectories, the encoder produces latent representations $Z_i$ summarizing the pre-treatment dynamics of each individual. Matching weights, including the reverse matching weights $(\hat{b}_{ji})_{j \in \mathcal{C}, i \in \mathcal{T}}$ introduced in the main text, are then learned exactly as in the synchronized setting. For a treated unit $i \in \mathcal{T}$, the counterfactual outcome is constructed as

$$\hat{Y}_{i,t}^0 = \sum_{j \in \mathcal{C}} \hat{b}_{ij} Y_{j,t}.$$

Thus, staggered adoption primarily modifies the representation learning stage through the masking process, while the matching mechanism itself remains unchanged.

**Fixed treatment time $t_0$.**    When all treated units receive treatment at the same time $t_0$, the estimator used in the main text is

$$\hat{\text{ATT}} = \frac{1}{n(L - t_0 + 1)} \sum_{t=t_0}^{L} \left[ \sum_{i \in \mathcal{T}} \left( \frac{\hat{g}(Z_i)}{\hat{P}_{11}} Y_{i,t} - \sum_{j \in \mathcal{C}} \hat{b}_{ij} \frac{\hat{g}(Z_j)}{\hat{P}_{10}} Y_{j,t} \right) + \sum_{j \in \mathcal{C}} \left( \sum_{i \in \mathcal{T}} \hat{b}_{ji} \frac{\hat{g}(Z_i)}{\hat{P}_{11}} Y_{i,t} - \frac{\hat{g}(Z_j)}{\hat{P}_{10}} Y_{j,t} \right) \right]. \tag{9}$$

Here $n = |\mathcal{T} \cup \mathcal{C}|$, and $\hat{P}_{11}$ and $\hat{P}_{10}$ are defined in Equation (4).

**Staggered treatment times.**    When treatment occurs at different times $t_i$ across treated units, the estimator can be adapted by averaging each treated unit over its own post-treatment window. This gives

$$\hat{\text{ATT}} = \frac{1}{n} \left[ \sum_{i \in \mathcal{T}} \frac{1}{L - t_i + 1} \sum_{t=t_i}^{L} \left( \frac{\hat{g}(Z_i)}{\hat{P}_{11}} Y_{i,t} - \sum_{j \in \mathcal{C}} \hat{b}_{ij} \frac{\hat{g}(Z_j)}{\hat{P}_{10}} Y_{j,t} \right) \right.$$
$$\left. + \sum_{j \in \mathcal{C}} \frac{1}{L - t_{\mathcal{C}} + 1} \sum_{t=t_{\mathcal{C}}}^{L} \left( \sum_{i \in \mathcal{T}} \hat{b}_{ji} \frac{\hat{g}(Z_i)}{\hat{P}_{11}} Y_{i,t} - \frac{\hat{g}(Z_j)}{\hat{P}_{10}} Y_{j,t} \right) \right]. \tag{10}$$

The only difference with the fixed-$t_0$ case is that each treated unit $i$ is evaluated from its own treatment time $t_i$. For the control contribution, we use a common reference time

$$t_{\mathcal{C}} = \max_{i \in \mathcal{T}} t_i,$$

so that control units are evaluated over a post-treatment period comparable across all treated units.

| ATT | Settings | | | |
|---|---|---|---|---|
| | (a) | (b) | (c) | (d) |
| True ATT | 1.54 | 1.54 | 1.54 | 1.98 |
| B-Twin ATT | (1.49,0.97) | (1.38,0.08) | ( 1.86,1.11) | (2.22,1.16) |

Table 3: ATT estimation on the second toy dataset with staggered treatment adoption. Reported values correspond to (mean ATT estimate, standard deviation) across 100 Monte Carlo simulations. The full time series length is $L = 168$, and treatment is introduced at $t_0 = 84$, $t_1 = 126$, and $t_2 = 142$, forming three cohorts of treated individuals.

**Interpretation.** The staggered extension therefore requires only two practical modifications: (i) Outcomes are masked using the treatment process before encoding, and (ii) post-treatment averages are computed relative to each treated unit's treatment time $t_i$. The latent matching mechanism, propensity estimation procedure, and counterfactual construction remain otherwise unchanged.

**Results.** Table 3 shows the results of B-Twin on the simulated dataset generated using the second simulation model, across the 4 settings given in Table 1. In this staggered setting, treated group is divided into three cohorts, with treatment times $t_0$, $t_1$ and $t_2$. The table reports the mean ATT estimate and standard deviation obtained across 100 Monte Carlo simulations for each setting. Overall, the results suggest that the proposed masking-based extension remains stable in this simplified staggered-treatment setting, with mean ATT estimates remaining close to the true ATT across all settings.

# B   Proof of Proposition 2

For completeness, we restate Proposition 2.

**Proposition 3** (Approximation of the latent propensity)**.** *Under the assumptions above, suppose that: (i) the reconstruction error satisfies*

$$\|Y^{\mathrm{pre}} - f_\theta(f_\phi(Y^{\mathrm{pre}}))\| \le \varepsilon_Y,$$

*and (ii) the propensity model satisfies*

$$|\hat{g}(Z) - \mathbb{P}(T = 1 \mid Z)| \le \varepsilon_P.$$

*Then,*

$$|\hat{g}(Z) - g(W)| \le \varepsilon_P + 2L_g\varepsilon_W + 2L_gL_h\varepsilon_Y.$$

*Proof.* **Step 1: Approximation of the latent propensity via reconstruction.**
Define $\hat{W} := h(f_\theta(Z))$. Then:

$$|g(W) - g(\hat{W})| \le L_g\|W - \hat{W}\|.$$

By the triangle inequality:

$$\|W - \hat{W}\| \le \|W - h(Y^{\mathrm{pre}})\| + \|h(Y^{\mathrm{pre}}) - h(f_\theta(Z))\|.$$

Using Assumption 4:

$$\|W - h(Y^{\mathrm{pre}})\| \le \varepsilon_W.$$

Using Lipschitz continuity of $h$ and the reconstruction error:

$$\|h(Y^{\mathrm{pre}}) - h(f_\theta(Z))\| \le L_h\|Y^{\mathrm{pre}} - f_\theta(Z)\| \le L_h\varepsilon_Y.$$

Thus:

$$\|W - \hat{W}\| \le \varepsilon_W + L_h\varepsilon_Y,$$

and therefore:

$$|g(W) - g(h(f_\theta(Z)))| \le L_g\varepsilon_W + L_gL_h\varepsilon_Y.$$

**Step 2: Approximation of the learned propensity.**
We decompose:

$$|\hat{g}(Z) - g(h(f_\theta(Z)))| \le |\hat{g}(Z) - \mathbb{P}(T = 1 \mid Z)| + |\mathbb{P}(T = 1 \mid Z) - g(h(f_\theta(Z)))|.$$

The first term is bounded by $\varepsilon_P$ from the propensity error. Define:

$$\varepsilon_Z := |\mathbb{P}(T = 1 \mid Z) - g(h(f_\theta(Z)))|.$$

Thus:

$$|\hat{g}(Z) - g(h(f_\theta(Z)))| \leq \varepsilon_P + \varepsilon_Z.$$

**Bounding the representation error term.**
Recall that

$$\varepsilon_Z = |\mathbb{P}(T = 1 \mid Z) - g(h(f_\theta(Z)))|.$$

Since $T \in \{0, 1\}$, we have:

$$\mathbb{P}(T = 1 \mid Z) = \mathbb{E}[T \mid Z].$$

Applying the law of iterated expectations:

$$\mathbb{E}[T \mid Z] = \mathbb{E}\big[\mathbb{E}[T \mid W, Z] \mid Z\big].$$

Since treatment assignment depends on the latent confounder $W$, and the representation $Z$ is constructed solely from pre-treatment outcomes generated through $W$, we have $T \perp\!\!\!\perp Z \mid W$. Therefore:

$$\mathbb{E}[T \mid W, Z] = \mathbb{E}[T \mid W] = g(W).$$

Therefore:

$$\mathbb{E}[T \mid Z] = \mathbb{E}[g(W) \mid Z].$$

Hence,

$$\mathbb{P}(T = 1 \mid Z) = \mathbb{E}[g(W) \mid Z].$$

Given this, we have:

$$\varepsilon_Z = |\mathbb{E}[g(W) \mid Z] - g(h(f_\theta(Z)))|.$$

Rewriting:

$$\varepsilon_Z = |\mathbb{E}[g(W) - g(h(f_\theta(Z))) \mid Z]|.$$

By Jensen's inequality:

$$\varepsilon_Z \leq \mathbb{E}\left[|g(W) - g(h(f_\theta(Z)))| \mid Z\right].$$

Using the previously established bound:

$$|g(W) - g(h(f_\theta(Z)))| \leq L_g \varepsilon_W + L_g L_h \varepsilon_Y,$$

we obtain:

$$\varepsilon_Z \leq L_g \varepsilon_W + L_g L_h \varepsilon_Y.$$

**Final bound.**
We combine:

$$|\hat{g}(Z) - g(W)| \leq |\hat{g}(Z) - g(h(f_\theta(Z)))| + |g(h(f_\theta(Z))) - g(W)|.$$

Using the previous bounds:

$$|\hat{g}(Z) - g(h(f_\theta(Z)))| \leq \varepsilon_P + L_g \varepsilon_W + L_g L_h \varepsilon_Y,$$
$$|g(h(f_\theta(Z))) - g(W)| \leq L_g \varepsilon_W + L_g L_h \varepsilon_Y,$$

we obtain:

$$|\hat{g}(Z) - g(W)| \leq \varepsilon_P + 2L_g \varepsilon_W + 2L_g L_h \varepsilon_Y.$$

$\square$

