# OpenReview forum: "Balanced Twins: Causal Inference on Time Series with Hidden Confounding"
_TMLR — Rejected by TMLR_

### Review · Reviewer_xGZc · 2026-03-22

**Summary Of Contributions:**

The authors propose a new method (called Balanced Twins) that improves counterfactual estimation in settings with latent confounding and when the data consists of time series. This counterfactual estimation is relevant for estimating the average treatment effect on the treated (ATT). The authors do this by first learning proxies of the latent confounder using a variational autoencoder. The authors then construct individual-level synthetic control using a novel balancing-constraint generalizing classical convexity constraints.

Main strengths:
- The authors consider a relevant problem and propose a novel method for estimating the ATT (as far as I can assess, as a disclaimer, I am only partially into the literature and might have overseen something)
- The authors provide extensive numerical (synthetic and real-world based) numerical simulations comparing their approach to several state-of-the-art methods.
- The authors provide code to reproduce their results



Main weaknesses:
- There seem to be quite some issues with indices. Sometimes your variables/potential outcomes have indices indicating time and individual, sometimes not. This also happens frequently within the same proof or equation (see further down below for details). I thus have some scepticism regarding the overall technical correctness of the claims, which apparently is the main criterion for TMLR.
- I find it difficult to follow which assumptions you make and which not. For example, it is stated that "in our work, ignorability does not hold"..., yet in the proof of Proposition 1, you invoke "ignorability". Stating your results and assumptions in there results (such as Proposition 1) more precisely, would help.

**Additional Comments:**

I have some other questions/comments:
- If your treatment $T_i$ might be administered at different time points, why is it not required for it to depend on the time index $t$?

**Audience:**

Yes

**Audience Explanation:**

I think this paper is of interest to some parts of the TMLR audience.

First, as far as I can assess, estimating causal effects (including the average treatment effect for the treated (ATT)) is a relevant topic and there is recent literature on comparable ML conferences on this topic.
The authors propose a new approach estimating the ATT, and this new approach shows promising numerical results, so therefore, I think this paper is of interest to parts of the TMLR community. However, I must add the disclaimer that I am only partially familiar with the literature and might have overseen something here.

**Broader Impact Concerns:**

The work is primarily methodological and located within the area of time series and causal effect estimation. Examples from the energy/electricity sector are discussed. I do not see any **specific** ethical concerns that need to be addressed here. If I see correctly, then the authors also have not provided such a statement, which, however, is not an issue I think as just argued.

**Claims And Evidence:**

Yes

**Claims Explanation:**

The authors provide extensive numerical simulations (synthetic and based on real-world data). The authors also provide code to reproduce these numerical simulations. The authors compare to relevant existing approaches, and by doing so, give evidence for the claims made about the limitations of existing work.

The authors also claim that their approach works on the considered hidden confounding setting, and the numerical simulations illustrate that it does.

I have more trouble with the mathematics: As explained above and in more detail below, there are issues with indices, how results and assumptions are stated, and thus, there are some issues that I think need to be fixed for this point.

However, as I have to condense this point into a Yes/No-decision, I am still leaning towards a Yes as I think the negative part can be fixed.

**Requested Changes:**

I will indicate which points are critical for my recommendation for acceptance and which I "only" think strengthen your work but are not that critical:
- (not critical) Some of your equations are properly punctuated, but many also aren't, see for example at the bottom of page 3. I think mathematical equations belong to the sentence in which they are introduced, in particular, if they end the sentence, they should be followed by a full stop.
- (critical) Below assumption 3 you write: "However, in our paper, ignorability does not hold ...". But then, in the proof of Proposition 1, you use some notion of ignorability. I think it would help if you clearly state the assumptions that you make (in particular, in Proposition 1). Do you refer to the assumption at the end of page 4 instead? Maybe give it a name, an equation reference? Why stating assumption 2 in an assumption environment, however, you then do not need it?
- (not critical) page 4: "In observational studies, we commonly make ..."-->"one commonly makes", as not all observational studies (I suppose) you are referring to are from you.
- (Critical) There are several issues with your indices that make it difficult to follow your paper and at the moment significantly reduce the quality of the paper: Some examples:   Sometimes your variables $Y^{(1)}$, $Y^{(0)}$, $T$ etc. are accompanied by indices sometimes not. Whether your assumptions are supposed to hold for all individual $i$ or just generically, is not clearly stated. What is $i$ in your assumption 3? An index for the individual, or something else? In Proposition 1 and your proof thereof, you frequently include or not include your indices that you use before. Sometimes you write $Y^{(0)}_{j,t}$, sometimes, you write $Y^{(0)}$, for example, for no apparent reason. It would also help if you include the index sets in the sums in proposition one to indicate over what you are summing.
- (Critical) You use different symbols for independence, sometimes with one bar (proof of Proposition 1), sometimes with two bars (Assumption 3).
- (not critical) on page 5, you also use the notation $\hat{Y}$ when introducing the SCM constraint, which has not been properly introduced.
- (not  critical) a subtlety: $g(W)$ or $g(w)$ is not a function, but $g(\cdot)$ is a function with argument $W$ or $w$
- (not critical) What are $q$ and $p$ on page 6 in the definitions of your loss(es), where have these been introduced?
- (not critical) I think you should properly introduce your abbreviations BCE and KL: While they appear in the text in the full names, without referring to these abbreviations though, the reader earlier has to guess what they mean (in particular, I would argue, for BCE)
- (critical) I think in particular in Section 4.6 you refer to several standard concepts without providing references. I think these should be included here.
- (not critical) page 16: "In this section, We" ---> "In this section, we"

---

### Review · Reviewer_2EcK · 2026-03-27

**Summary Of Contributions:**

In this work, the authors focus on the estimation of a treatment occurring in time series at different times for each unit, in the presence of unobserved confounding. The authors propose an approach to learn a latent representation of the pre-treatment trajectories, that would enable further adjustment on unobserved confounders, allowing to obtain a counterfactual trajectory of treated units using a variant of synthetic control, and hence to estimate the ATT. They provide experiments on simulated, semi-synthetic and real data to show the performances of their method, in comparison to other approaches, and their limits.

**Audience:**

Yes

**Audience Explanation:**

Causal inference on time series is a very broad topic. Even though the performances of B-Twin are not entirely convincing, the benchmark is interesting and a good guide for the practitioner to decide which method they should use.

**Broader Impact Concerns:**

no concern

**Claims And Evidence:**

Yes

**Claims Explanation:**

The method is clearly presented, and the experiments show adequate performances in comparison with other methods. B-Twin is clearly exposed and motivated. However, B-Twin performs better than competitors only in a very particular setting that should be further motivated.

**Requested Changes:**

- The performance claims of the article should be toned down, and the strong case of the method should be better explained. B-Twin only works better in the case where there is a distribution shift. The plausibility of such a setting in real data should be further discussed and illustrated with examples. The authors claim that a difficulty of the setting they consider is that the treatment starts at different times for all units, which makes systematic shifts (like a punctual event) less likely to affect all units at the same time of the treatment application. Overall, if I were to estimate a treatment effect on such data after reading the article, I would not use B-Twin, which seems slower and less performant than other methods except in a setting which seems very unlikely, so the authors should focus on a more convincing use case explanation
- The description of the problem is very succint and insufficient. Figure 1 should illustrate the whole time series, as well as different treatment times for different units, and so on. A very simple textbook DAG showing only a confounder X (a notation which is introduced but not used in the framework, until a remark in paragraph 4.2 p.6 which should intervene much earlier in the writing) is a little disappointing, and does not help a unfamiliar reader understand the contribution. This is crucial for acceptance.
- there is no limitations paragraph
- it would be interesting to discuss the relation of the considered framework to the regression discontinuity setting

---

### Review · Reviewer_3Dz2 · 2026-04-27

**Summary Of Contributions:**

This paper proposes B-Twin, a neural synthetic-control-style method for estimating treatment effects in time-series settings with hidden confounding and staggered treatment adoption. The method learns a latent representation from pre-treatment time series, estimates a propensity-like score from this representation, and then constructs individual counterfactuals as weighted combinations of control trajectories.

The problem is important, and the paper is right to emphasize that hidden confounding, nonstationarity, and staggered adoption make naive comparisons unreliable. A strength of the paper is its attempt to retain the interpretability of synthetic control while scaling the construction to individual-level high-frequency time series.

However, the main weakness is fundamental. The paper’s theoretical balancing condition is stated in terms of g(W)=P(T=1|W), where W is hidden, but the implemented method learns g_hat(Z) from a learned representation Z. The paper does not justify when g_hat(Z) is a valid substitute for g(W). This breaks the connection between the theoretical balancing condition and the algorithm. As a result, the current paper reads more like a neural, individual-level synthetic-control heuristic than a convincing causal method for hidden confounding.

**Audience:**

No

**Audience Explanation:**

No. The problem area is relevant to TMLR, but I do not think the current findings are reliable enough for the audience to benefit from them.

The paper addresses an important setting: time-series treatment effect estimation with hidden confounding and staggered treatment adoption. However, the main finding rests on an unjustified substitution. The theoretical condition is written in terms of g(W)=P(T=1|W), where W is hidden, but the algorithm learns g_hat(Z) from a learned representation Z. Since the paper does not establish when g_hat(Z) is a valid substitute for g(W), the claimed balancing argument does not follow.

As a result, the empirical findings are difficult to interpret causally. They may show that the proposed neural synthetic-control-style estimator performs well on the authors' simulations and benchmarks, but they do not establish that the method solves hidden confounding in the stated causal setting. Without a valid bridge from the theoretical object g(W) to the implemented object g_hat(Z), the paper's main message would likely mislead rather than inform TMLR readers.

**Broader Impact Concerns:**

The paper uses privacy-sensitive ICU and household electricity-consumption data, so it should explicitly address data-use terms, de-identification, and privacy risks. There is also potential harm if an unvalidated hidden-confounding method is used for clinical or energy-policy decisions, especially for vulnerable patients or households.

**Claims And Evidence:**

No

**Claims Explanation:**

The main claims are not supported by a clear causal identification argument.

The most serious problem is the transition from the theoretical balancing condition to the implemented algorithm. In Proposition 1, the paper defines the propensity function as g(w)=P(T=1|W=w), where W is the hidden confounder (p. 5). The same page then says that "we cannot balance on W directly" and proposes to use "g(W) as a proxy" in the balancing condition (Eq. 3, p. 5). But if W is unobserved, then g(W) is also unobserved. This is not an observed proxy; it is still a function of the hidden variable.

The implementation does not estimate g(W) directly. Instead, the algorithm uses a learned representation Z and a classifier head g_hat(Z). On p. 6, the paper defines g_hat(Z_i)=sigma(g_psi(Z_i)) and trains it with L_p=BCE(T,g_hat(Z)). This objective learns a treatment predictor from Z, i.e. something related to P(T=1|Z), not the theoretical quantity P(T=1|W). The paper does not state an assumption under which P(T=1|Z)=P(T=1|W), nor does it prove that Z is sufficient for the treatment-relevant information in W.

This is not a minor technical gap; it is an impossibility without additional assumptions. To see this, suppose W is binary and hidden, while the learned representation Z is constant, say Z=0 for all units. Consider two data-generating models. In Model A, P(W=1)=1/2, P(T=1|W=0)=0.2, and P(T=1|W=1)=0.8. In Model B, P(W=1)=1/2, P(T=1|W=0)=0.5, and P(T=1|W=1)=0.5. Both models induce exactly the same observed treatment probability conditional on Z: P(T=1|Z=0)=0.5. Therefore, any classifier trained on (Z,T), including the paper’s BCE(T,g_hat(Z)) head, sees the same learning problem in both worlds. Yet the theoretical propensity functions g(W)=P(T=1|W) are completely different: Model A has g(0)=0.2 and g(1)=0.8, while Model B has g(0)=g(1)=0.5.

Thus g(W) is not identifiable from Z and T in general. No neural architecture, VAE, classifier head, or balancing loss can recover g(W) from Z unless the paper adds an assumption that Z preserves the treatment-relevant information in W. The current manuscript does not state such an assumption. Consequently, the move from Eq. (3), which is written in terms of g(W), to the implemented loss involving g_hat(Z), is not justified. The proposed balancing objective may be a predictive heuristic, but it is not a valid causal balancing argument for hidden confounding as written.

There are also formulation issues around staggered adoption. The paper says that "Treatment can be initiated at different times" (Problem Formulation, p. 3), but the formal setup mainly uses a binary treatment indicator T_i in {0,1}. If staggered adoption is central, the SCM should define an adoption time S_i and a time-indexed treatment process A_{i,t}. A binary T_i only says whether a unit was treated during the observed period; it does not encode when treatment begins.

Finally, the role of covariates X is unclear. The problem formulation defines time-indexed covariates X_i={x_{i,t}}_{t=1}^L (p. 3), and Figure 1 presents X as affecting both treatment and outcome. However, the method section later says the experiments use "pre-treatment time series as input" and that additional covariates X can be concatenated (p. 6). If X is part of the causal adjustment structure, it cannot be treated as an optional architectural add-on. The paper should clarify whether X is a required observed confounder, an optional predictive feature, or absent from the empirical instantiations of the method.

Overall, the central theoretical object in the paper is g(W), but the algorithm optimizes g_hat(Z). The paper never bridges this gap. As a result, the main causal claim collapses: the method is not shown to balance hidden confounding, and the evidence provided does not establish the claimed identification logic.

**Requested Changes:**

Critical change required for acceptance:

The paper must be rewritten around a valid causal identification argument. The current approach is not a minor fix away from being sound. The theoretical balancing condition is written in terms of g(W)=P(T=1|W), where W is hidden, but the algorithm replaces it with g_hat(Z), learned from a representation Z via BCE(T,g_hat(Z)). Without a stated and defensible assumption that Z preserves the treatment-relevant information in W, g(W) is not identifiable from Z and T. Therefore the proposed balancing condition cannot support the paper's claim to address hidden confounding.

A revision would need to restart from the causal problem formulation: define the SCM, the treatment timing/adoption process, the observed data, the target estimand, and the assumptions under which the missing counterfactual is identified. Only after that should the authors propose an estimator. If the authors cannot justify the transition from the hidden object g(W) to an observed or learned quantity, then the causal claim should be removed and the paper should be reframed as a predictive synthetic-control-style heuristic rather than a method for causal inference under hidden confounding.

Non-critical suggestions if the paper is rewritten:

- Clarify the role of staggered adoption with an adoption time S_i and treatment path A_{i,t}.
- Clarify the role of covariates X and whether they are required confounders or optional predictive inputs.
- Reframe the novelty as neural individual-level synthetic control rather than a new identification strategy.
- Tone down real-data claims where ground truth treatment effects are unavailable.

---

### Author Response · Authors · 2026-05-10

We thank all reviewers for the detailed and constructive feedback. Following the reviews, we substantially revised the manuscript to strengthen the theoretical justification, clarify the assumptions and notation, clarify the discussion of the staggered treatment formulation, and better position the scope and limitations of the method.

The main revision concerns the theoretical connection between the latent propensity \(g(W)=P(T=1\mid W)\) and the learned propensity \(\widehat g(Z)\). We agree with Reviewer 3 that the original manuscript did not sufficiently justify under which conditions \(\widehat g(Z)\) can approximate \(g(W)\). To address this issue, we now introduce two additional assumptions in Section 3 (“Problem Formulation”). In particular, we introduce an explicit latent recoverability assumption stating that the treatment-relevant component of the hidden confounder can be approximately recovered from pre-treatment trajectories. Under this assumption, together with Lipschitz regularity assumptions on the recovery map and propensity function, we prove a new approximation result bounding the error between \(\widehat g(Z)\) and \(g(W)\) (see Proposition 2 in Section 4, and full proof in Appendix B):
\[|\widehat g(Z)-g(W)| \le \varepsilon_P + 2L_g\varepsilon_W + 2L_gL_h\varepsilon_Y.\]

The error depends on how informative the pre-treatment trajectories are about the treatment-relevant hidden confounder, the reconstruction error, and the propensity score estimation error.
The revised manuscript therefore no longer relies on the assumption that arbitrary hidden confounding can be addressed from outcome trajectories alone. Instead, the causal interpretation is now explicitly restricted to settings where hidden confounding leaves a sufficiently informative signature in the observed pre-treatment trajectories. We also added a dedicated limitations section discussing this assumption and the corresponding failure modes.

Following Reviewer 2’s comments, we substantially revised the motivation of the paper and added the limitations paragraph. The revised manuscript now better explains that B-Twin is not intended as a universally superior alternative to outcome-regression approaches, but rather as a complementary approach designed for settings with latent confounding and non-stationary outcome dynamics where extrapolation from pre-treatment trajectories may be unreliable. We expanded the discussion of realistic examples of global temporal shifts and unit-level non-stationarity in the introduction (Section 1). We also revised the DAG and problem formulation to highlight the presence of the hidden confounder affecting pre-treatment outcomes, post-treatment outcomes, and treatment participation. We also clarified that the role of covariates \(X\) is optional, since when they are available and induce confounding, they can be used as conditioning variables and fed to the encoder to construct the latent representations \(Z\). Finally, although we believe standard Regression Discontinuity assumptions are not naturally satisfied in our setting because there is no observed running variable and no known cutoff determining treatment probability, we added Regression Discontinuity in Time (RDiT) as an additional baseline for the simulated experiments, where intervention time serves as the temporal running variable.

Following Reviewer 1’s suggestions, we also revised the problem formulation and notation throughout the paper. In particular, we clarified the distinction between generic random variables and indexed individual/time-dependent quantities, standardized the notation used in the proofs and clarified the assumptions used throughout the paper.
We also clarified the staggered treatment setting. In the revised manuscript, \(T_i\) only indicates whether a unit belongs to the treated or control population, while treatment onset times may differ across individuals through treatment times \(t_i\). The experiments use a common treatment onset time \(t_0\) primarily for simplicity and comparability with existing baselines. We additionally explain in the appendix how B-Twin extends to staggered settings using treatment-process masking:
\[A_{i,t}=\mathbf{1}\{t \ge t_i\},\]
where the encoder receives both \(Y_{i,t}\) and \(A_{i,t}\), and representation learning is performed on masked outcome trajectories \(Y_{i,t}(1-A_{i,t})\). We also added additional staggered-treatment experiments in the appendix to illustrate the proposed masking-based extension in a simplified staggered setting with multiple treatment cohorts.These experiments are intended as a proof-of-concept validation of the masking strategy rather than a comprehensive evaluation over all possible staggered-adoption regimes. (see Section 3 and the appendix for more details).
Finally, we corrected notation inconsistencies, properly introduced abbreviations, fixed equation punctuation, added missing references, and revised the overall presentation for clarity.

---

> ### Comment · Reviewer_xGZc · 2026-05-18
> **Reply**
>
> I suppose I am reviewer 1.
>
> I thank the authors for the new version of the paper and the suggested changes. I think the paper is in much better shape now. Only a few remaining points:
>
> - page 4: $Y^1_t$ is not introduced at this point in the ATT-formula, only later in section 4 you introduce it. I think you can solve that issue by writing out $Y^1_i$ and $Y^0_i$ as you did for $Y_i$ on the same page.
>
> - page 4: superfluous empty space in front of of "However, in our setting ..."
>
> - I like that you stated your assumptions much more clearly now. Still, maybe slightly confusing: Assumptions 2 and 3 still lack an individual-index i; maybe, if you do not want to include this index at this point, write that you understand these assumptions to hold for all $i$. (This only becomes clear somewhat later in Section 4, but not at this point necessarily).
>
> - sometimes you use \eqref to refer to equations, sometimes \ref, but this is not a too important point I think
>
> - Isn't there an index $i$ still missing in equation (1) for the treatment variable $T$? Similarly, for equation (3)?
>
> - I think a reference for the Adam-optimizer is missing

---

> ### Author Response · Authors · 2026-05-20
>
> Thank you very much for the careful reading and the detailed feedback.
> We will address all these points in the next revision. In particular, we will clarify the notation for (Y_i^1) and (Y_i^0), fix the remaining inconsistencies, add the missing indices, make the equation references consistent throughout the paper, and add the missing reference for the Adam optimizer.
> Thank you again for the constructive suggestions.

---

> > ### Comment · Reviewer_xGZc · 2026-05-20
> > **Reply**
> >
> > Thanks for further incorporating these points.
> >
> > The points I made in my original review are thus addressed. I am not an competent enough to comment on the discussion with Reviewer 3; for my final decision I will hence focus on the points from my review yet state that I am somewhat uncertain with my decision.

---

### Decision · Action_Editor_ksE3 · 2026-06-03

**Recommendation:** Reject

**Additional Comments:**

Following Reviewer 3Dz2's comments (see below), a major revision should rebuild the causal argument from the problem formulation upward. The central revision needed is to justify, weaken, or reframe the latent recoverability assumption. The authors should also provide either observable diagnostics, sensitivity analysis, or a more limited claim that does not rely on unverifiable recovery of W.

If resubmitted, the authors should also complete the reproducibility materials, including clear instructions for obtaining/preprocessing the real datasets and making the final code available in a standard repository.

### Reviewer 3Dz2's final justification:

The revision does not resolve my main concern. It assumes away the problem.

The new latent recoverability assumption states that there exists a function h such that W can be approximately recovered from Y^{pre}: ||W - h(Y^{pre})|| <= epsilon_W.

This is exactly the missing bridge in the original manuscript. The paper does not derive this condition, does not learn h, does not estimate epsilon_W, and does not provide any observable diagnostic for whether the condition holds. Since W is unobserved, the assumption cannot be checked from the data, and there are no ways to learn such function
.

The paper’s stated goal is to learn a \hat g such that \hat g(Z) is close to g(W). But W is unobserved, so the training data never contain the target g(W). The model is trained with BCE(T, \hat g(Z)), which can at best learn P(T=1 | Z), not P(T=1 | W). These are different objects unless Z is already sufficient for W with respect to treatment assignment. That sufficiency is exactly what the paper needs to justify. It cannot be obtained merely by training a treatment classifier on Z.

Also, the new bound |\hat g(Z) - g(W)| <= epsilon_P + 2 L_g epsilon_W + 2 L_g L_h epsilon_Y is useful only if epsilon_W is small. But epsilon_W is precisely the unobserved error in recovering the hidden confounder. If W is not recoverable from Y^{pre}, the bound is vacuous. The paper gives no empirical or theoretical reason to believe epsilon_W is small in the real settings motivating the paper.

This is a very strong proxy-confounding assumption. It is not a general method for hidden confounding. It is a method for the special case where the hidden confounder is already encoded in the observed pre-treatment trajectory strongly enough to be approximately recovered. That is a much narrower problem.

There is also still a gap between propensity approximation and causal estimation. Even if \hat g(Z) were close to g(W), the paper still needs to show that the learned weights induce the required balance for the counterfactual outcome and control the ATT estimation bias. Proposition 2 only bounds a surrogate propensity error. It does not establish that the learned weights balance the latent confounding structure, nor that the resulting counterfactual estimator identifies the ATT.

Thus the revision masks the original issue with an untestable and very strong assumption. The central causal claim remains unsupported.

**Audience:**

Yes

**Audience Explanation:**

The combination of representation learning, propensity-style balancing, and synthetic-control-style counterfactual construction is likely to be of interest, and the empirical comparisons may be useful to readers working on causal estimation for temporal data.

**Claims And Evidence:**

No

**Claims Explanation:**

The submission addresses an important problem, and the empirical section appears substantially improved, with two reviewers judging the revision positively. The main unresolved issue is the bridge between a theoretical object and the implemented model. The revision introduces a latent recoverability assumption, but this effectively assumes the key difficulty away. Moreover, the revised bound is only meaningful when the unobserved recoverability error is small, and the manuscript does not provide a convincing empirical or theoretical reason to believe this in the motivating settings.

**Resubmission Of Major Revision:**

The authors may consider submitting a major revision at a later time.